# A cell-type deconvolution meta-analysis of whole blood EWAS reveals lineage-specific smoking-associated DNA methylation changes

Chenglong You[1], Sijie Wu[1,2,3], Shijie C. Zheng [1,4], Tianyu Zhu [1], Han Jing[1], Ken Flagg[5], Guangyu Wang[6], Li Jin[1,2,3], Sijia Wang [1,7 ✉] & Andrew E. Teschendorff [1,8 ✉]

Highly reproducible smoking-associated DNA methylation changes in whole blood have been reported by many Epigenome-Wide-Association Studies (EWAS). These epigenetic alterations could have important implications for understanding and predicting the risk of smoking-related diseases. To this end, it is important to establish if these DNA methylation changes happen in all blood cell subtypes or if they are cell-type specific. Here, we apply a cell-type deconvolution algorithm to identify cell-type specific DNA methylation signals in seven large EWAS. We find that most of the highly reproducible smoking-associated hypomethylation signatures are more prominent in the myeloid lineage. A meta-analysis further identifies a myeloid-specific smoking-associated hypermethylation signature enriched for DNase Hypersensitive Sites in acute myeloid leukemia. These results may guide the design of future smoking EWAS and have important implications for our understanding of how smoking affects immune-cell subtypes and how this may influence the risk of smoking related diseases.

[1] CAS Key Laboratory of Computational Biology, CAS-MPG Partner Institute for Computational Biology, Shanghai Institute of Nutrition and Health, Shanghai Institute for Biological Sciences, University of Chinese Academy of Sciences, Chinese Academy of Sciences, 320 Yue Yang Road, Shanghai 200031, China. [2] Human Phenome Institute, Fudan University, 825 Zhangheng Road, Shanghai, China. [3] State Key Laboratory of Genetic Engineering and Ministry of Education Key Laboratory of Contemporary Anthropology, Collaborative Innovation Center for Genetics and Development, School of Life Sciences, Fudan University, Shanghai, China. [4] Department of Biostatistics, Johns Hopkins Bloomberg School of Public Health, Baltimore, MD, USA. [5] Guangzhou Regenerative Medicine Guangdong Laboratory, Guangzhou, China. [6] Department of Computer Science and Technology, Tsinghua University, Beijing, China. [7] Center for Excellence in Animal Evolution and Genetics, Chinese Academy of Sciences, Kunming 650223, China. [8] UCL Cancer Institute, Paul O'Gorman Building, University College London, 72 Huntley Street, London WC1E 6BT, UK. ✉email: wangsijia@picb.ac.cn; andrew@picb.ac.cn

Alterations in DNA methylation (DNAm) that accrue in normal cells as a function of age and exposure to environmental factors have been proposed to play a critical role in disease development[1,2]. Among the various phenotypes and exposures, aging and smoking stand out as two where highly reproducible DNA methylation changes have been identified[3–9]. Smoking-associated DNAm changes have attracted particular attention: for instance, such DNAm changes in blood have been shown to correlate with future lung cancer risk and incidence[10–12], with all-cause mortality[13,14], and with health span[15,16]. However, whether these DNAm changes are causally implicated in smoking-related pathologies like lung cancer or cardiovascular disease is still unclear and a matter of debate[10,17,18].

In order to better understand the role of smoking-associated DNAm changes in smoking-related disease etiology, it is important to determine the cell-type specificity of such changes in tissues like blood or lung[19,20]. Indeed, relatively little is known about the cell and tissue-type specificity of smoking-associated DNA methylation changes, in contrast to aging where most of the DNAm changes appear to be independent of tissue and cell-type[21]. So far, most epigenome-wide-association studies (EWAS) in smoking have been performed in easily accessible but heterogeneous tissues like whole blood[7,22–29] or buccal swabs[9]. A number of small-scale studies have been performed in highly relevant cell-types like lung macrophages[30–32], but were generally underpowered to assess if smoking-associated DNAm changes are common across different cell-types. Other studies have focused on tumor-adjacent normal colonic mucosa[33], or tumor-adjacent normal lung tissue[34], which have identified smoking-associated DNAm changes shared with blood, yet did not completely adjust for the high level of immune-cell infiltration: approximately 40% of cells in lung tissue are immune cells[35,36], which could confound analyses and interpretation. Thus, putative smoking-associated DNAm signatures derived in such solid tissues may reflect the alterations already known to be present in the blood cells that infiltrate the tissue. Indeed, we recently confirmed this for the case of buccal swabs, a tissue that consists mainly of squamous epithelial and immune-cell infiltrates[35,37–39] by demonstrating that known smoking-associated DNAm changes in blood are fully recapitulated in the immune-cell compartment of buccal swabs from smokers[40].

Focusing only on the immune-system, it is equally unclear whether smoking-associated DNAm changes reported in whole blood EWAS occur simultaneously across all immune-cell subtypes or only in specific subsets. This is a critically important question given the fundamental role the immune response and inflammation plays in the development of cancer[18,41,42] and cardiovascular disease[43,44]. Some EWAS performed in purified blood cell populations have begun to address this important question, but were limited to one or two cell-types[45], or limited in terms of sample size[46,47]. Thus, preliminary findings from these studies await confirmation from large EWAS performed in purified blood cell subtypes, ideally encompassing all major cell subtypes. Such studies would ideally also derive the different blood cell subtype samples from the same individuals, yet performing such a multi cell-type EWAS in a large cohort is extremely labor-intensive and costly. An alternative much cheaper strategy is to apply a computational algorithm designed to extract cell-type specific differential DNAm signal from an EWAS performed in whole blood[40]. A number of such algorithms have recently been proposed[40,48–50]. Although the full extent to which these algorithms can carry out proper inferences at cell-type resolution on real data is still unclear, one of these algorithms called CellDMC was validated across several different real EWAS[40], suggesting that inference of cell-type specific smoking-associated DNAm changes may be possible. Indeed, CellDMC is able to identify cell-type specific differential methylation by incorporation of statistical interaction terms in the linear model, which can resolve variation in effect-size as a function of cell-type abundance in a cell-type specific manner[40].

Here, we apply CellDMC to 7 independent large EWAS cohorts, totaling 4448 samples, encompassing 2 main ethnicities (Chinese and White Caucasian) and tissue types (blood and buccal swabs). Consistent across all seven studies, we find that the highly reproducible smoking-associated hypomethylation signature in blood, as summarized in Gao et al.[8], is largely specific to cells within the myeloid lineage, encompassing mainly neutrophils and monocytes. A meta-analysis over the seven studies further reveals a novel smoking-associated hypermethylation signature in myeloid cells and a few loci specifically altered in lymphoid cells. Thus, our study provides a novel gold-standard list of smoking-associated differentially methylated cytosines at the highest yet cellular resolution.

## Results

**Smoking-associated DNAm changes in a Chinese cohort.** Previous studies have established that smoking associated DNAm changes are largely shared between cohorts of European and African ancestry[28]. We decided to test this for a Chinese population, in order to justify its inclusion in a subsequent meta-analysis. We performed Illumina DNA methylation profiling at over 850,000 CpGs[51] in whole blood from a total of 712 Chinese individuals (after QC) (the "TZH" cohort, "Methods"). DNAm data were normalized according to a standard protocol that we have validated many times before[52], including BMIQ for type-2 probe bias correction[53] and COMBAT to adjust for beadchip effects[54], which were evident in an SVD-analysis[55]. Blood cell type fractions were estimated using our EpiDISH algorithm and DNAm reference matrix defined over seven blood cell subtypes[56] ("Methods"). These fractions were in line with those commonly observed in blood EWAS, with neutrophils defining the major component (Supplementary Fig. 1). Of the 712 samples, a total of 688 reported lifetime smoking habits, which included 453 never-smokers, 62 ex-smokers, and 173 smokers at sample draw (Supplementary Data 1). We identified smoking-associated differentially methylated cytosines (DMCs) by performing ordinary linear regression analysis with DNAm as the response and smoking as the exposure, whilst also adjusting for potential confounders including age, sex, array-position and blood cell type fractions. We note that smoking was encoded as an ordinal variable (0 = never smoker, 1 = ex-smoker, 2 = current smoker) and was chosen over smoking-pack-years, because the latter was not available for all the cohorts to be included in the later meta-analysis. We observed a relatively strong association between smoking and DNAm, with 417 smoking-associated DMCs passing an false-discovery rate (FDR) < 0.05 threshold (Fig. 1a). Previously derived gold-standard lists of smoking-associated CpGs exhibited the expected patterns of DNAm change in the TZH cohort. For instance, a gold-standard list of 62 smoking-associated CpGs derived from a review of 12 blood EWAS[8], exhibited significant association in our Chinese cohort (Fig. 1a), with the great majority of these displaying hypomethylation, as observed previously[8]. The fact that the top hits (e.g., cg05575921 at the *AHRR* locus), are exactly as those in most previous smoking EWAS, demonstrates that these smoking-associated DMCs are largely independent of ethnicity. Another more comprehensive list of 2622 smoking-associated CpGs, which includes the above 62 sites and which was derived from a meta-analysis over 16 cohorts (Joehanes et al.[28]), also exhibited the same directional smoking-associated DNAm changes in our TZH cohort (Fig. 1b, Fisher-test $P < 1e-100$). This strong association

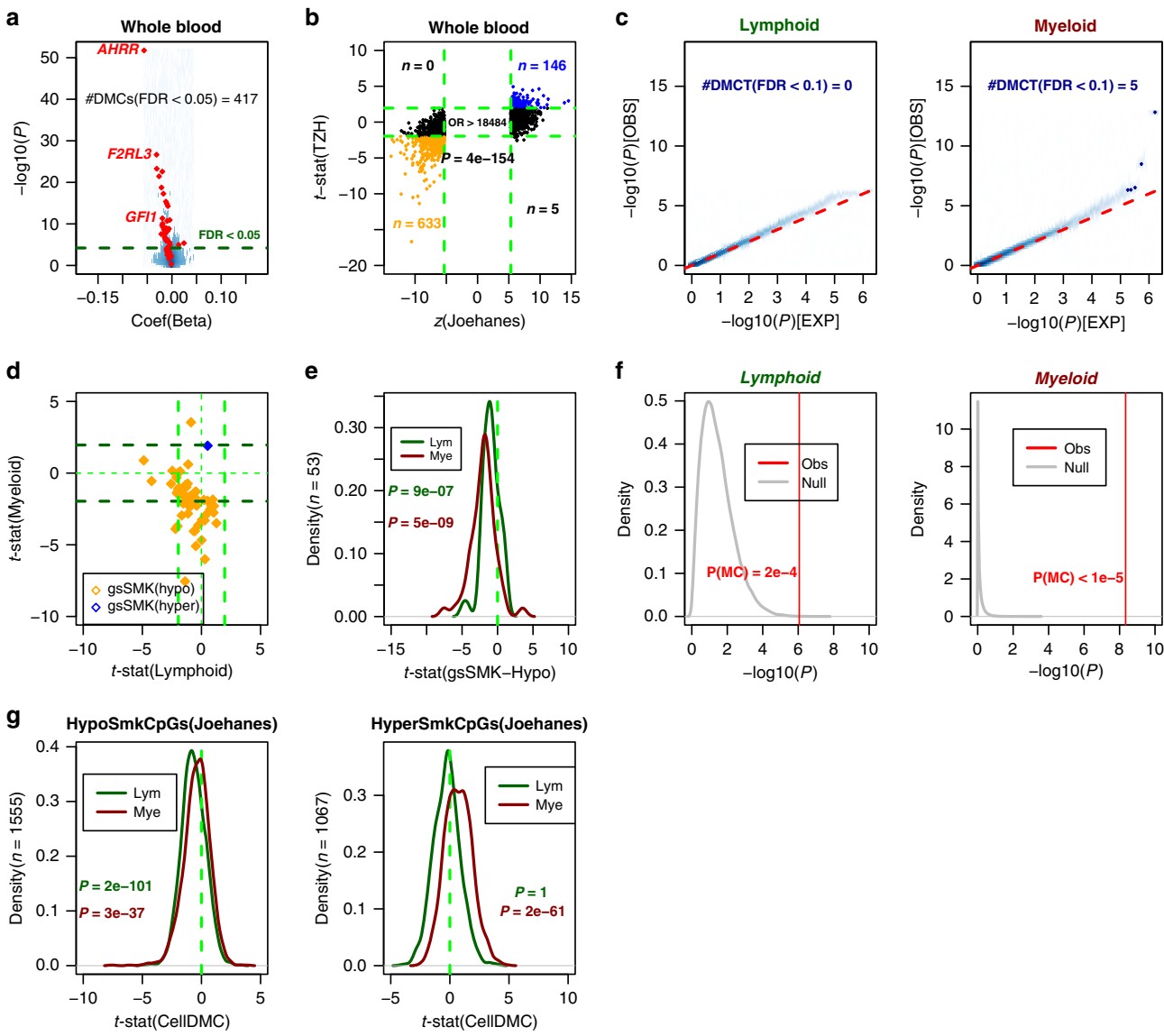

**Fig. 1 Blood lineage specific smoking-associated DNAm changes in TZH cohort. a** Volcano scatterplot for 811,902 CpGs displaying the estimated coefficient ($x$-axis) and $-\log_{10}P$ value ($y$-axis) derived from a linear model of DNAm against smoking status. Dashed line denotes FDR = 0.05, and in red we highlight the set of 62 gold-standard smoking-associated CpGs. The number of DMCs passing FDR < 0.05 is given. **b** Scatterplot of $z$-statistics of the 2622 smoking-associated CpGs from the meta-analysis of Joehanes et al. ($x$-axis) vs. their corresponding $t$-statistics in the TZH cohort. Odds ratio (OR) and one-tailed Fisher-test $P$ value are given. Consistently hypermethylated and hypomethylated CpGs are shown in blue and orange, respectively. Vertical and horizontal green dashed lines represent the Bonferroni and FDR < 0.05 significance levels in the Joehanes and TZH, respectively. The number of CpGs in each significant quadrant are indicated by "$n$". **c** Quantile–quantile plots displaying the results of CellDMC for the lymphoid and myeloid cell-types, as indicated. Red dashed line indicates the line where observed $P$ values match the expected ones under the null-hypothesis of no global association between smoking and differential DNAm in the respective cell-type. Blue data points represent the DMCTs passing an FDR < 0.3 threshold. **d** Scatterplot of CellDMC-derived $t$-statistics of smoking-associated differential DNAm in lymphoid cell-type ($x$-axis) vs. the corresponding $t$-statistic in myeloid cell-type ($y$-axis). Dashed lines represent the $P = 0.05$ thresholds, and $t = 0$. In orange (blue) we depict the gold-standard smoking-associated CpGs that passed QC and that are hypomethylated (hypermethylated) in smokers. **e** Density distributions of CellDMC-derived $t$-statistics of smoking-associated differential DNAm in the TZH cohort for 53 gold-standard smoking hypomethylated CpGs in the lymphoid and myeloid lineages, as indicated. $P$ value derives from a one-tailed Wilcoxon test, testing that these gold-standard CpGs do exhibit a significant trend toward hypomethylation. **f** Monte-Carlo significance analysis of hypomethylation trends for the 53 gold-standard hypomethylated smoking-CpGs. Density plots of the $-\log_{10}P$ values derived from a one-tailed Wilcoxon rank sum test to determine if the $t$-statistics of association between DNAm and smoking of 53 randomly selected CpGs is significantly less than zero, for a total of 100,000 Monte-Carlo runs ("Null", gray). The vertical red line indicates the corresponding one-tailed $P$ value for the 53 gold-standard hypomethylated smoking-CpGs from Gao and Brenner. $P$ values in red are derived from the Monte-Carlo analysis. All the data in this figure derives from analysis in our Chinese cohort. **g** Density distributions of CellDMC-derived $t$-statistics of smoking-associated differential DNAm in the TZH cohort for the 2622 gold-standard smoking CpGs from Joehanes et al in the lymphoid and myeloid lineages, as indicated. The 2622 CpGs have been split according to hypo (left) or hypermethylation (right) in the Joehanes meta-analysis study. $P$ values derive from a one-tailed Wilcoxon test, testing that these gold-standard CpGs do exhibit a significant trend towards hypo-or-hypermethylation, as required.

was driven by 146 consistently hypermethylated and 633 consistently hypomethylated CpGs, with the remaining CpGs not exhibiting significant changes (Fig. 1b).

**CellDMC reveals distinct myeloid and lymphoid smoking DNAm signatures.** Next, we applied our CellDMC algorithm[40] to the TZH cohort data in order to determine the specific cell-types driving the observed smoking-associated differential methylation. CellDMC runs the same linear model as before, but now including statistical interaction terms between phenotype (i.e., smoking status at sample draw) and cell-type fractions. Although EpiDISH estimates fractions for seven blood cell subtypes (B-cells, NK, CD4+, and CD8+ T-cells, monocytes, neutrophils, and eosinophils), we decided to run CellDMC at a more coarse-grained level that assumes only two major cell types, i.e., myeloid (monocytes, neutrophils + eosinophils) and lymphoid cells (B-cells, T-cells, and NK-cells). This was done in order to more robustly identify separate lymphoid and myeloid smoking-associated DNAm signatures ("Methods"). However, at an FDR < 0.05 threshold, CellDMC predicted no DMCTs, indicating lack of power. Relaxing the FDR threshold to 0.1, we obtained 5 myeloid DMCTs (Fig. 1c), with these numbers increasing to 50 myeloid and 35 lymphoid DMCTs if relaxed further to FDR < 0.3.

Next, we explored the CellDMC predictions for the gold-standard lists of 62 smoking-associated DMCs (henceforth gsSMK-CpGs) from Gao and Brenner[8] and the 2622 gsSMK-CpGs from Joehanes et al.[28]. Of the 62 gsSMK-CpGs, 60 have been observed to undergo hypomethylation in smokers in at least 3 independent whole blood EWAS[8], of which 54 were present on the EPIC beadarray, with 53 of these passing QC. We observed that 62% of these 53 CpGs exhibited hypomethylation in the myeloid cells of smokers, albeit not always with genome-wide statistical significance (Fig. 1d). Only one of the two hypermethylated gsSMK-CpGs is present on the EPIC beadarray, and this one was marginally hypermethylated in the myeloid cells (Fig. 1d). The same direction of effect was observed in the lymphoid cells, albeit not as strong as in the myeloid compartment (Fig. 1d) in line with the observed reduced global significance levels (Fig. 1c). To determine whether the trend towards hypomethylation in each lineage is statistically significant, we performed a Wilcoxon rank sum test (Fig. 1e), supplemented by a Monte-Carlo analysis (Fig. 1f), to ascertain that the skew toward hypomethylation could not have arisen by random chance. These analyses confirmed that the 53 gsSMK-CpGs exhibited highly significant trends toward hypomethylation in smokers in both lineages, but more strongly so in myeloid cells (Fig. 1e, f).

Next, we investigated the pattern of DNAm for the other gold-standard list of 2622 CpGs from Joehanes et al. We split the list into the 1555 and 1067 that exhibited hypo-and-hypermethylation in their meta-analysis, respectively. Interestingly, whilst for the 1555 hypomethylated gsSMK-CpGs we observed a statistically significant trend towards hypomethylation in both myeloid and lymphoid lineages, for the 1067 hypermethylated gsSMK-CpGs a corresponding trend toward hypermethylation was only seen in the myeloid lineage (Fig. 1g).

**Validation in three independent whole blood cohorts.** We sought to validate the above findings for the gsSMK-CpGs in independent large whole blood EWAS. To this end we applied CellDMC to Illumina 450k DNAm datasets consisting of 689 whole blood samples from Liu et al.[57], a separate cohort of 656 whole blood samples from Hannum et al.[6], and 464 blood samples from Tsaprouni et al.[25], all with available smoking exposure information ("Methods", Supplementary Data 1). After normalization and adjustment for potential confounders ("Methods"),

CellDMC predicted DMCTs in both lymphoid and myeloid lineages, but consistency was only observed for the gsSMK-CpGs in the myeloid lineage (Fig. 2a). In line with this and the results in the TZH cohort, the 60 hypomethylated gsSMK-CpGs exhibited a clear trend toward hypomethylation in smokers in the Liu, Hannum, and Tsaprouni cohorts, which was only consistent and much stronger in myeloid cells (Fig. 2b–d). Indeed, in Liu the trend toward hypomethylation in the lymphoid lineage was only evident when assessing all 60 gsSMK-CpGs globally, with only 1 CpG exhibiting statistical significance (FDR < 0.05), in stark contrast to the myeloid lineage where 34 of the 60 gsSMK-CpGs exhibited statistically significant (FDR < 0.05) hypomethylation (Fig. 2b). Similar patterns were seen in Tsaprouni and Hannum, with marginal or no evidence of hypomethylation in the lymphoid lineage, in stark contrast to the myeloid cells (Fig. 2c, d). These data suggest that the well-known and highly reproducible smoking-associated hypomethylation at gsSMK-CpGs in whole blood is mainly driven by corresponding hypomethylation in myeloid cells.

While the validation of the top ranked DMCTs in the myeloid lineage is reassuring, the discrepancy between the cohorts in the lymphoid lineage motivated us to seek additional confirmation of our results. First, we compared our findings to that of a recent study (Su et al.) which performed a small-scale EWAS in 5–6 purified blood cell subtypes[46], identifying blood cell subtype specific smoking-associated DNAm changes. In particular, they focused on a panel of 7 CpGs mapping to *AHRR*, *ALPPL2*, *GFI1*, *IER3*, *F2RL3*, *GPR15*, and *ITGAL*, for which DNAm had been measured in smokers and non-smokers in CD14+ monocytes, CD15+ granulocytes, CD19+ B-cells, and CD2+ T-cells[46]. This study found that *AHRR*, *ALPPL2*, *GFI1*, *IER3*, and *F2RL3* were hypomethylated predominantly or almost specifically in neutrophils and monocytes (i.e., myeloid cells), whereas *GPR15* and *ITGAL* were specifically hypomethylated in lymphoid cells. We thus compared the CellDMC predictions for these seven CpGs across the four independent whole blood cohorts, which revealed strong consistency with Su et al. (Fig. 3). Indeed, cg19859270 (*GPR15*) was predicted by CellDMC to be hypomethylated predominantly in lymphoid cells, whereas the five CpGs mapping to *AHRR*, *ALPPL2*, *GFI1*, *IER3*, *F2RL3* were correctly predicted to be hypomethylated predominantly in the myeloid lineage (Fig. 3). The only CpG not consistent with Su et al. was cg09099830 mapping to *ITGAL*, which according to Su et al. should be hypomethylated only in lymphocytes.

**Further validation in HIV samples and buccal swabs.** As further validation of our results, we ran CellDMC on three additional cohorts, profiled with Illumina 450k/850k DNAm beadarrays ("Methods"). Two of these cohorts consist of 608 and 529 blood samples (Zhang et al.[58]), both from HIV patients (Supplementary Data 1). Blood cell subtype proportions in these HIV cohorts were similar to those of the other cohorts, albeit with a marginally higher lymphoid and marginally lower myeloid fraction (Supplementary Fig. 2). We further verified that adherence to anti-retroviral therapy (ART) did not impact blood cell subtype proportions (Supplementary Figs. 3 and 4). Application of CellDMC to these HIV cohorts also revealed highly consistent patterns to those observed in blood of healthy individuals (Supplementary Fig. 5). The other third cohort consists of 790 buccal swab samples from women all aged 53[9]. Buccal swabs are known to contain both squamous epithelial cells as well as immune cell infiltrates, thus for these samples we applied HEpiDISH[35] to first infer total epithelial, total lymphoid and total myeloid fractions in each sample (Supplementary Fig. 6). In the buccal swabs, CellDMC revealed that the 60 hypomethylated gsSMK-CpGs exhibited a clear trend towards hypomethylation in the myeloid

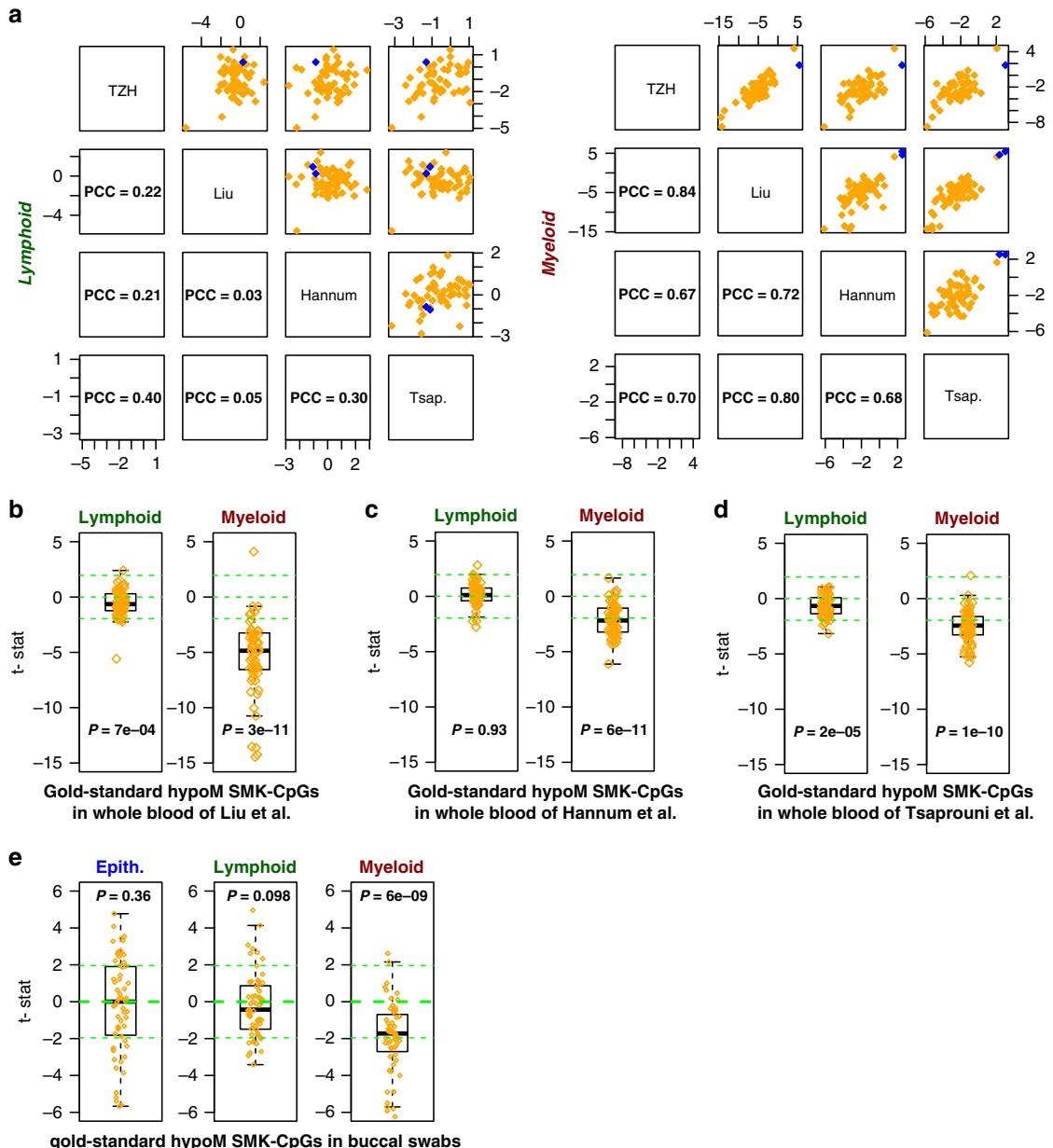

**Fig. 2 Cell-type specific smoking-associated DNAm patterns of gold-standard smoking CpGs in independent cohorts. a** Scatterplots of cell-type specific *t*-statistics (as obtained with CellDMC) across four whole blood cohorts (TZH, Liu, Hannum, and Tsaprouni) with associated Pearson Correlation Coefficient (PCC) values, and separately for the lymphoid (left) and myeloid (right) lineages. The *t*-statistics are displayed for the 62 gold-standard smoking-associated CpGs (60 hypomethylated (orange) and 2 hypermethylated (blue)). **b** Boxplots of *t*-statistics of association of DNAm with smoking in the whole blood (*n* = 689) from Liu et al., as derived with CellDMC for both lymphoid and myeloid lineages. Boxplots only display the 60 gold-standard smoking hypomethylated CpGs from Gao and Brenner. *P* value derives from a one-tailed Wilcoxon rank sum test. Horizontal line within boxes indicate median, box-boundaries the interquartile range, and whiskers extend to 1.5 times this range. **c–e** As **b**, but now for the 656 samples from Hannum et al., the 464 samples from Tsaprouni et al., and the 790 buccal swab samples.

lineage, but not so in the epithelial or lymphoid lineages (Fig. 2e), consistent with the findings in blood.

Next, we explored the patterns of association for the same panel of seven CpGs considered earlier, and which have been tested for smoking-associated DNAm changes in purified FACS-sorted blood cell subtypes. This revealed strong consistency with the TZH, Liu, Hannum, and Tsaprouni cohorts, as well as with Su et al., i.e the five CpGs mapping to *AHRR*, *ALPPL2*, *GFI1*, *IER3*, *F2RL3*, which according to Su et al. are hypomethylated specifically (or predominantly) in myeloid cells, were correctly predicted by CellDMC to be hypomethylated specifically in the myeloid cells present in buccal swabs (Fig. 3), and myeloid cells of HIV patients

(Supplementary Fig. 7). Similarly, of the two CpGs mapping to *GPR15* and *ITGAL*, which according to Su et al. are predominantly hypomethylated in lymphoid cells, CellDMC correctly predicted this pattern for both CpGs in buccal swabs (Fig. 3), while for the HIV-cohorts results were in line with those seen in the blood samples of healthy cohorts (Supplementary Fig. 7).

**Meta-analysis reveals novel myeloid smoking-associated DNAm signatures.** In order to increase power, we next performed a genome-wide meta-analysis over the seven cohorts in order to determine whether an extended smoking DNAm

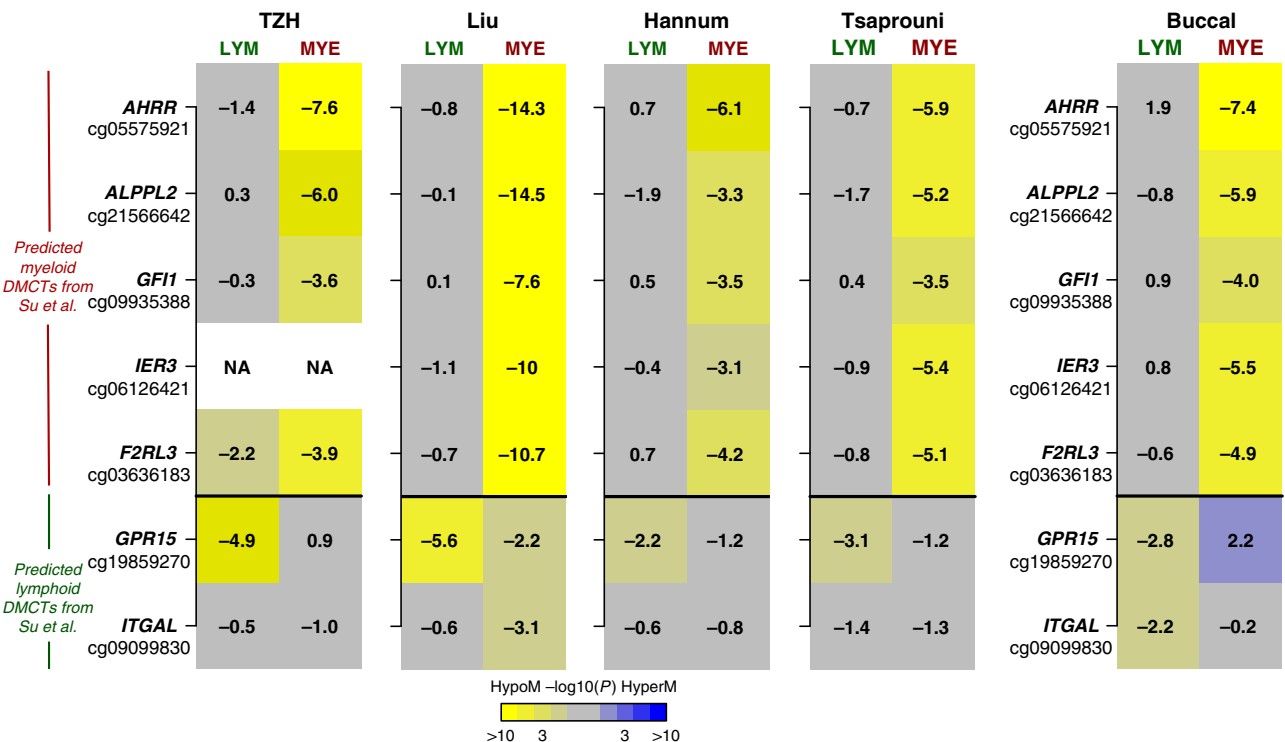

**Fig. 3 Validation of CellDMC's predictions.** Heatmaps of myeloid and lymphoid significance *P* values, as derived from CellDMC, in five separate cohorts and for a panel of seven CpGs which Su et al. showed to exhibit myeloid and lymphoid specific hypomethylation in smokers. The first four cohorts are whole blood samples. Significance of *P* values is denoted by color, and the corresponding *t*-test statistic values are displayed in the heatmap. *P* values derive from the *t*-test in the CellDMC model and are two-tailed.

signature specific to each hematopoietic lineage exists ("Methods"). We used Efron's empirical Bayes approach in combination with the Stouffer's meta-analysis method[59,60] to obtain aggregate meta-analysis *P* values for a common set of 356,158 CpGs from the CellDMC-derived statistics in each cohort ("Methods"), which we subsequently adjusted for multiple testing using Benjamini–Hochberg (BH) procedure[61]. FDR estimates were broadly consistent with those derived from an empirical permutation procedure ("Methods", Fig. 4a). At an FDR < 0.05 threshold, this revealed 173 DMCTs in the myeloid lineage, but only 1 DMCT (cg19859270, *GPR15*) in lymphoid cells. Among the 173 myeloid DMCTs, 103 were consistently hypomethylated across at least 6 of the 7 studies, whereas 67 were consistently hypermethylated (Fig. 4b). Of the 62 gsSMK-CpGs from Gao and Brenner, 48 were assessed in the meta-analysis, of which 37 were among our 173 myeloid DMCTs, with 36 of these in the hypomethylated group and one (cg12803068) mapping to *MYO1G* in the hypermethylated group. Using a more relaxed threshold (FDR < 0.3) a total of 536 myeloid DMCTs and 4 lymphoid DMCTs (Fig. 4b, Supplementary Datas 2 and 3) were detected, which included 42 of the 48 assessed gsSMK-CpGs from Gao & Brenner. At the same FDR threshold, 76 and 75% of the 536 myeloid and 4 lymphoid DMCTs were found in the list of 2622 gsSMK-CpGs from Joehanes et al.[28], with the overlapping CpGs all exhibiting the same directional DNAm change between our meta-analysis and that of Joehanes et al. (Supplementary Data 4).

In order to gain insight into the nature of the 536 myeloid DMCTs, we used the eFORGE tool[62,63], which tests for enrichment of cell-type specific DNase hypersensitive sites (DHS). This analysis was performed for DHS profiled in specific blood cell subtypes as generated by BLUEPRINT[64], and carried out separately for the 250 consistently hypomethylated and 252

consistently hypermethylated myeloid DMCTs. For the hypomethylated category the strongest enrichment was for DHS in inflammatory macrophages, which we note is consistent with the smoking-associated DMCTs occurring in the myeloid lineage (Fig. 4c). Among the implicated genes (Supplementary Data 5), it is worth highlighting *ZEB2*, a transcription factor that has recently been shown to play a key role in monocyte and macrophage identity[65,66], as well as retinoic acid receptor alpha (*RARA*), which plays a key role in the maintenance of immune homeostasis during inflammatory responses[67], and which has been implicated in aortic dissection[44]. Interestingly, for the hypermethylated myeloid DMCTs, the only observed enrichment was for DHS as defined in acute myeloid leukemia (AML) (Fig. 4c). This is intriguing given that AML is the only hematological cancer for which smoking is a major risk factor[68,69]. Among the implicated genes (Supplementary Data 6) it is worth highlighting *RAD52*, an enzyme involved in DNA repair of double-strand breaks, *TELO2*, a regulator of DNA damage response kinases like *ATM*[70,71], and *RPTOR*, an essential component of the mTORC1 complex which has been implicated in AML development and progression[72].

Given that our meta-analysis only revealed 4 lymphoid-DMCTs, enrichment analysis is not possible. Besides cg19859270 (*GPR15*), cg02657160 (*CPOX*) was also hypomethylated in lymphoid cells across all 7 studies, whilst the other two (cg09837977, *LRRN3* & cg08529529, *ALOX5AP*) were hypomethylated in six. Of note, two of the implicated genes (*GPR15* & *LRRN3*) have been shown to be associated with smoking in a recent transcriptome-wide meta-analysis performed in whole blood[73]. *LRRN3* in particular was also identified to be one of the few genes exhibiting simultaneous DNAm and gene-expression changes in blood in association with smoking[74].

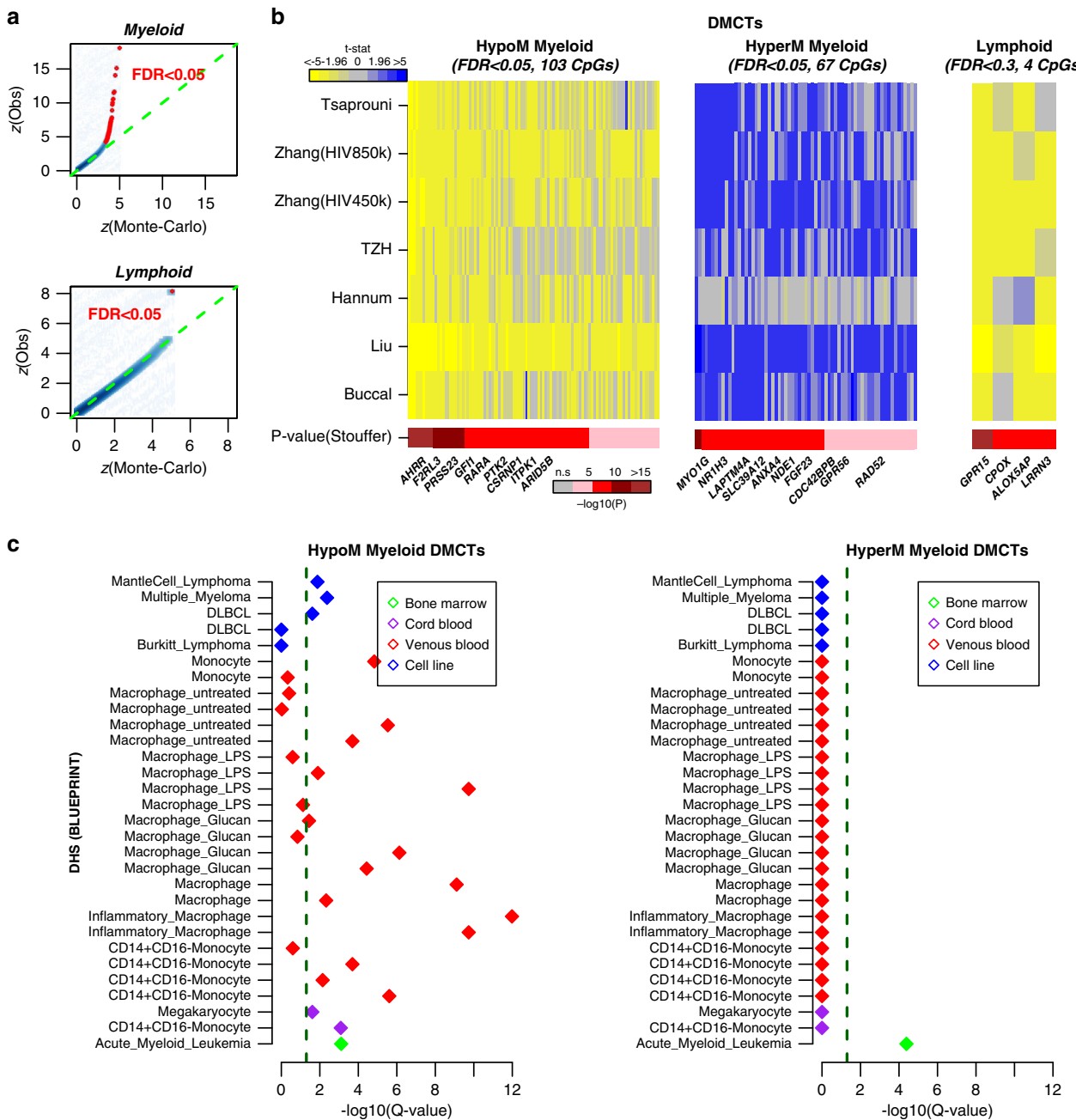

**Fig. 4 Myeloid and lymphoid DMCTs as revealed by meta-analysis. a** Smoothed scatterplots of the expected quantile of the absolute *z*-statistic in the meta-analysis, as inferred from a Monte-Carlo permutation analysis (*x*-axis) vs. the observed quantile (*y*-axis). CpGs that passed an FDR < 0.05 thresholds are shown in red. **b** Heatmaps of CellDMC *t*-statistics for myeloid and lymphoid DMCTs, that were deemed statistically significant in a meta-analysis over the seven studies, as indicated by the Stouffer-test *P* value. For the myeloid lineage, we used a FDR < 0.05 threshold and only display DMCTs which were consistently hypomethylated or consistently hypermethylated across at least six of the seven studies. For the lymphoid lineage, an FDR < 0.3 threshold was used. **c** eFORGE enrichment analysis results on the myeloid hypomethylated and hypermethylated DMCTs. The *y*-axis labels the cell-types for which enrichment of DNase Hypersensitive Sites (DHSs) among the hypo-and-hypermethylated DMCTs was tested. *x*-axis labels −log₁₀(*q*-value), where the *q*-value is the FDR estimate. The green dashed line indicates the significance threshold FDR < 0.05. Nature of cell-types is labeled by color, as indicated. DLBCL diffuse large B-cell lymphoma, LPS lipopolysaccharide.

**Simulation model predicts few lymphoid-specific DMCTs**. Our meta-analysis, as well as the analysis in the individual cohorts, suggests a scarcity of smoking-associated lymphoid-DMCTs. In order to determine whether this result reflects a lack of power or underlying biology, we devised a simulation model ("Methods") to estimate the expected sensitivity of CellDMC to detect three separate categories of DMCTs: (i) non-specific DMCTs that occur in both myeloid and lymphoid lineages, (ii) lymphoid-specific DMCTs (i.e., DMCTs present in lymphoid cells but not present in myeloid cells), and (iii) myeloid-specific DMCTs (i.e., DMCTs present in myeloid cell but not present in lymphoid cells). For the simulation, we used realistic cell-type fractions derived from the TZH cohort (mean myeloid fraction = 72%, mean lymphoid fraction = 28%). We also estimated realistic effect sizes for

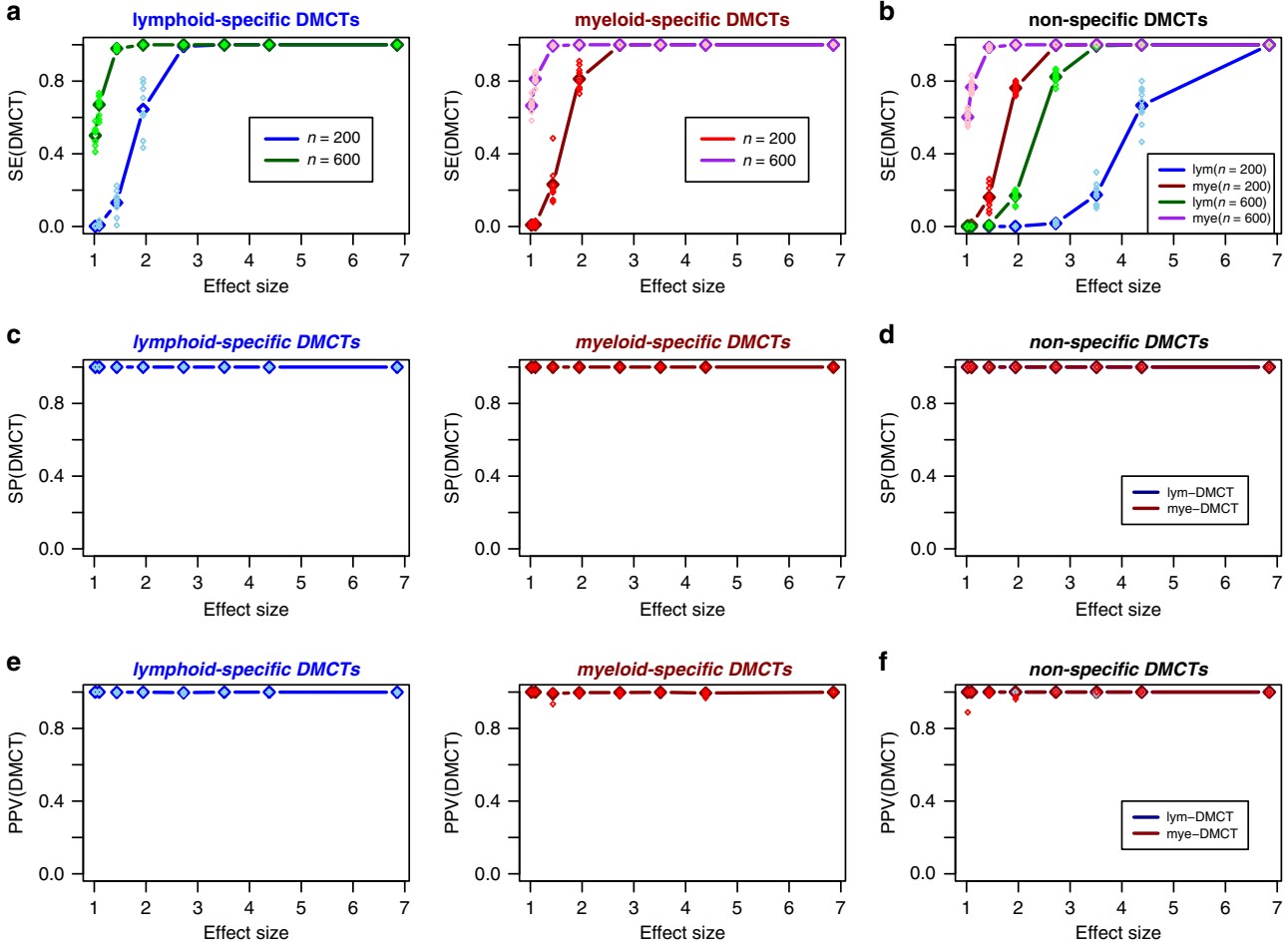

**Fig. 5 Simulation-based power estimates for detecting lymphoid and myeloid DMCTs. a** Plot of the sensitivity to detect lymphoid and myeloid specific DMCTs ($y$-axis) vs. the smoking effect size ($x$-axis). Data points represent the mean sensitivity, as obtained over 10 different Monte-Carlo simulations, encompassing 482,077 CpGs of which 1000 are DMCTs. Darkblue/darkred datapoints are for $n = 200$ (100 cases and controls), darkgreen/purple are for $n = 600$ (300 cases and controls). Each sample is an in silico mixture of real DNAm profiles representing one purified CD4+ T-cell and one monocyte sample. For each case sample, 1000 DMCTs at the given average effect size in only one of the two cell-types was generated. The sensitivities for each of the ten Monte-Carlo runs are shown in skyblue/red. **b** As **a**, but now for the scenario where the DMCTs are introduced at the same CpG in both cell-types. Sensitivity to detect the DMCT in the lymphoid and myeloid lineage is shown. **c**, **d** As **a**, **b**, but now for the specificity. **e**, **f** As **a**, **b**, but now for the precision or positive predictive value (PPV). Specificity and PPV values for both choices of sample size are effectively identical.

myeloid and lymphoid cells, derived from the observed effect sizes in whole blood and the estimated cell-type fractions ("Methods"). Assuming an average effect size in individual cell-types of approximately 1.6 (corresponding to a mean methylation difference of about 10%, "Methods"), and a total sample size of 600 (e.g., 300 smokers and 300 never-smokers), we have estimated a sensitivity of over 90% to detect lymphoid-specific and myeloid-specific DMCTs respectively (Fig. 5a). However, for DMCTs occurring simultaneously in both lineages, the sensitivity to detect them in the lymphoid lineage is severely compromised, while the sensitivity to detect them in the myeloid lineage remains over 90% (Fig. 5b). Thus, unless effect sizes are large, DMCTs that occur simultaneously in myeloid and lymphoid cells would be hard to detect. We also estimated the precision (PPV) to correctly detect DMCTs, as well as the specificity, defined as one minus the FPR, which quantifies the proportion of false positive calls among all true non-DMCTs ("Methods"). In line with our previous study[40], specificity and PPV were always close to 1 (Fig. 5c–f, Supplementary Fig. 8), which is consistent with our observations on real data. Overall, the obtained results support the view that lymphoid-specific DMCTs are rare.

## Discussion

By applying CellDMC[40] to seven large independent EWAS cohorts, we have here performed virtual in silico blood cell-type specific EWASs, thus benefiting from the high sample numbers in these studies, whilst simultaneously avoiding the labor-intensive effort and high economic cost of profiling multiple purified blood cell subtypes in such large numbers of individuals. Although CellDMC was extensively tested and validated on both simulated and real EWAS data in our previous work[40], it is important to stress that sensitivity is limited, especially if one seeks to identify DMCTs at high cellular resolution. There are approximately seven major blood cell subtypes and we found that application of CellDMC at this level of resolution exhibited limited sensitivity across specific cell-types (data not shown). To overcome this, we ran CellDMC at the resolution of lymphoid and myeloid lineages, or in the case of buccal swabs, at the resolution of 3 main cell-types (epithelial, lymphoid, and myeloid cells). While running CellDMC at this coarser level is a limitation, the ability to infer smoking-associated DNAm changes that may be specific or common to the two major hematopoietic lineages is a novel question that is of paramount interest. Indeed, to the best of our

knowledge, this study is among the first to perform such a virtual hematopoietic lineage-specific EWAS, and thus one of the first studies to comprehensively assess whether the prominent and highly reproducible smoking-associated DNAm signature seen in whole blood, as summarized by Gao and Brenner[8], is present in both myeloid and lymphoid lineages.

Our analysis in the TZH cohort as well as the meta-analysis over the 7 EWASs encompassing 4448 samples has revealed that the 62 CpG smoking signature from Gao and Brenner is more prominent in the myeloid lineage. This result is very much line with that of previous preliminary studies addressing the cell-type specific nature of smoking-DMCs. For instance, Su et al.[46] performed a small-scale case/control study (around 20 cases and controls) in purified granulocytes, monocytes, B-cells and T-cells, focusing on a panel of well-known smoking-associated loci as determined by whole blood EWAS. Although this study was limited in terms of sample size and number of loci examined, they did show that some of the well-known smoking-associated loci like AHRR, F2RL3, GFI1, and GPR15, exhibit blood cell-type specific changes, in broad agreement with findings from another small-scale smoking-EWAS[47], thus justifying the use of these loci as a means to test the validity of CellDMC's predictions. Indeed, according to CellDMC across the seven cohorts analyzed here, the panel of five myeloid-specific loci considered in Su et al. (AHRR, F2RL3, GFI1, ALPPL2, IER3) exhibited consistent and significant hypomethylation in myeloid cells, but not so in lymphoid cells, whilst the lymphocyte-specific marker GPR15 exhibited the reverse pattern. That gsSMK-CpGs derived from whole blood are more prominent in the myeloid lineage is also supported by another study that assessed smoking associated DNAm changes in lymphoblasts and pulmonary macrophages[30]. This study only identified one marginal association in lymphoid cells in comparison to many highly significant hits found in macrophages[30]. Another study by Reynolds et al[45] assessed smoking-associated DNAm changes using Illumina 450k data in a large cohort of purified monocyte samples, and although it could not assess specificity, it did demonstrate that most of the gsSMK-CpGs exhibit corresponding smoking-associated DNAm changes in monocytes[45].

Importantly, our lineage-specific meta-analysis performed over the seven studies also revealed a number of novel insights. First, it has demonstrated that the smoking-associated hypomethylation signature in myeloid cells is highly enriched for DHS as defined in inflammatory macrophages. This could have important ramifications for understanding the effect of smoking on inflammatory processes within organs like the lung or heart. For instance, a well-known gene in this myeloid-specific hypomethylation signature is AHRR, the repressor of the aryl-hydrocarbon-receptor (AHR) detoxification pathway[75]. The AHR pathway is activated by xenobiotic toxic chemicals (e.g., aromatic polycyclic hydrocarbons) in cigarette smoke and is thought to play a key role in metabolizing them[76]. The AHR-pathway also plays an important tumor suppressive role in lung inflammation[77]. The inactivation of the AHR-pathway, possibly initiated and mediated by hypomethylation and overexpression of AHRR, could be an important early step in lung cancer development[18]. A novel gene present in the myeloid hypomethylation signature and which merits further discussion is ZEB2, which controls monocyte and macrophage identity[65,66]. It is plausible that smoking-associated disruption of ZEB2 regulatory activity could skew lung alveolar macrophage polarization to an inflammatory phenotype that promotes tumor development[30]. RARA was also part of the myeloid hypomethylation signature, and given its role in immune-system homeostasis and inflammation[67], its deregulation could be important in cardiovascular disease. For instance, RARA differential methylation has been observed in the context of aortic dissection, for which smoking is a major risk factor[44].

Second, our meta-analysis has revealed a myeloid-specific smoking-associated 63-CpG hypermethylation signature, which was found to be enriched for DHS as profiled in AML. This is noteworthy, as smoking is a well-known risk factor for AML[68,69], and not a risk factor for lymphomas or multiple myeloma, for which corresponding DHS were not enriched. Some of the genes (e.g., RPTOR) in this signature are also relevant to AML development[72], including RAD52 and TELO2 which have key roles in DNA repair[70,71]. The presence of such DNA repair enzymes is important because tobacco smoke is well-known to induce many double-strand breaks, which in turn are known to recruit DNMT1 and EZH2 to these sites[78]. Indeed, a recent study performed in bronchial epithelial cells has demonstrated how exposure to cigarette smoke can lead to repressive polycomb (EZH2) marking and hypermethylation at many developmental genes, sensitizing cells to oncogenic transformation[78]. Thus, the myeloid-specific hypermethylation signature identified here may represent an analogous process occurring in myeloid progenitor cells present in the bone marrow. Exploring the relevance of this smoking-associated hypermethylation signature in AML will be an interesting task for future studies.

Third, our meta-analysis has revealed that there are few consistent alterations happening in the lymphoid compartment. In fact, only cg19859270 (GPR15) passed a meta-analysis FDR < 0.05 threshold, thus confirming the observation in Su et al. that this locus is lymphocyte-specific, but also demonstrating this specificity across 7 large cohorts encompassing over 4400 samples. Relaxing the FDR threshold to FDR < 0.3, only revealed an additional three lymphoid-specific DMCTs. As remarked earlier, one of these mapped to LRRN3, a gene found associated with smoking in a recent transcriptome-wide meta-analysis performed in whole blood[73], and also one of the few genes exhibiting simultaneous DNAm and gene-expression changes in blood in association with smoking[74]. Our analysis further suggests that smoking-associated DNAm changes at LRRN3 are specific to lymphocytes, which is an entirely novel insight. Of note, another recent study has shown that both GPR15 and LRRN3 exhibit elevated expression in the blood of smokers that have had an ischemic stroke, for which smoking is a risk factor[79]. Among the other lymphoid-specific DMCTs, cg02657160 (CPOX) has previously been reported to be specifically hypomethylated in peripheral blood mononuclear cells (PBMCs) of smokers, but not in whole blood[47]. Since PBMCs are devoid of granulocytes and enriched for lymphocytes, this is consistent with our meta-analysis result. Thus, given that LRRN3 and CPOX have been previously implicated as exhibiting DNAm alterations in smokers, this lends support to our statistical significance estimates. Finally, it is worth highlighting the lymphoid-DMCT mapping to ALOX5AP, a gene implicated in the leukotriene pathway, and with polymorphisms that have been associated with increased risk of coronary artery disease[80]. Synergistic interactions between single-nucleotide polymorphisms (SNPs) in this gene with cigarette smoking have also been reported in relation to an elevated risk of atherosclerotic cerebral infarction[81]. Of note, Su et al. reported another CpG that exhibited lymphocyte-specific smoking-associated DNAm changes (mapping to ITGAL), which however, did not change significantly in our meta-analysis, except for a marginal association in the buccal swab study. It follows that either ITGAL is a false-positive finding of Su et al., or CellDMC could not detect ITGAL, perhaps due to the fact that ITGAL only exhibited significant hypomethylation in CD8+ T-cells, in contrast to the observed hypomethylation of GPR15 which was significant in both T- and B-cells[46]: a DNAm alteration present in both T-cells and B-cells is more likely to be picked out by CellDMC, as such an alteration is effectively present in most lymphocytes, whereas a change occurring in B-cells but not T-cells would be much harder to detect.

There are a number of plausible explanations as to why our analysis has detected relatively few DMCTs in the lymphoid lineage. Biologically, it could reflect the fact that the majority of lymphoid cells (i.e., T-cells) undergo further development and maturation in the thymus. Thus, in contrast to monocytes and neutrophils, which are produced in the bone marrow, it is plausible that smoking-associated changes present in precursor T-cells within the bone marrow are erased when they undergo further development into mature T-cells in the thymus. While lymphoid-specific smoking-associated changes could also be acquired in the thymus, this also seems to be less likely. The absence of many DMCTs in the lymphoid lineage could, however, also be due to a lack of power. Indeed, we stress that our results need to be interpreted with great caution, because the fraction of lymphocytes in whole blood is lower (i.e., around 30% compared to a myeloid fraction of 70%), and also less variable between healthy individuals, compared to that of neutrophils. Supporting this, the simulation analysis performed here has shown that whilst the sensitivity to detect myeloid-specific and lymphoid-specific DMCTs is reasonably high, that the sensitivity to detect non-specific DMCTs in the lymphoid lineage is compromised by the fact that the myeloid proportion in whole blood is much higher. Thus, it is possible that a proportion of myeloid-DMCTs are not truly myeloid-specific, exhibiting changes also in the lymphoid lineage, and that CellDMC just lacked power to detect the changes in lymphoid cells. On the other hand, we also stress that the comparison to previous studies (e.g., Su et al.), as discussed earlier, suggests that this lack-of-power may not be an issue for the top ranked myeloid-DMCTs, which do seem to be myeloid-specific. In future, it will be interesting to apply algorithms like CellDMC to thousands of samples merged together in one dataset, as this may improve power.

In conclusion, the analyses and data presented here support the view that most smoking-associated DNAm changes reported in whole blood are driven by corresponding changes in myeloid cells. This may help guide the design of future smoking EWAS, but may also have key implications and ramifications for our understanding of smoking-related disease etiology.

## Methods

**Illumina DNAm EPIC dataset (TZH cohort)**. Blood samples from three Chinese cities (Zhengzhou, Taizhou, Nanning) were sent to Fudan University Taizhou Institute of Health Sciences for storage at $-80\,^{\circ}\text{C}$ until DNA extraction. Henceforth, we refer to this cohort as the TZH cohort. The TZH cohort study was conducted with the official approval of the Ethics Committee of the Shanghai Institutes for Biological Sciences (ER-SIBS-261410). The Declaration of Helsinki Principles was followed and all participants provided written informed consent. DNA extraction was performed using a TGuide M48 Automated nucleic acid extractor. Genome-wide DNA methylation was profiled in the Infinium MethylationEPIC BeadChips (Illumina). Five hundred nanogram of genomic DNA from each whole blood sample was bisulfite converted using the EZ DNA Methylation Kit (Zymo Research). BeadChips were processed following the manufacturer guide and protocol for Infinium MethylationEPIC array. DNA was hybridized to BeadChips and single base extension were performed using a Freedom EVO robot (Tecan). BeadChips were subsequently imaged using the iScan Microarray Scanner (Illumina). Illumina.idat files were then processed with the *minfi* Bioconductor package[82] without background correction (although background correction reduces bias it does so at the expense of increased variance, which is generally something to be avoided, unless the DNAm data are used for copy-number estimation). Probes with SNPs were removed using the *dropLociWithSnps* function from *minfi*.

This function uses the SNP information provided by Illumina and UCSC Common SNP tables (including version 132, 135, 137, 138, 141, 142, 144, 146, and 147) with preset MAF (0 is the default value and was used here) to filter SNP CpGs. We further removed probes on chromosomes X and Y. We further used the Illumina definition of $\beta$ values and derived $P$ values of detection for the rest of probes by comparing the total intensity U + M to that of the background distribution (given by negative control probes), as implemented in *minfi*. $\beta$ values with $P$ values of detection greater than 0.01 were set to NA. Of note, the threshold of detection ($P < 0.01$) is more stringent than the $P < 0.05$ threshold used in the other cohorts, partly because sample coverages were very high, allowing for a more

stringent threshold while also retaining a high coverage over probes. Only probes with less than 5% missing values were retained. The missing $\beta$ values were then imputed with the impute.knn function (using $k = 5$) in R. Type-2 probe bias was corrected using BMIQ[53]. All this resulted in a 811,902 probe times 712 sample data matrix. Based on principal component analyses, we found a significant slide/beadchip effect. Therefore we used ComBat[54] on $M$-values (logit of $\beta$ values) to correct for the slide effect and then transformed the $M$-values back to $\beta$ values.

**Independent smoking EWAS cohorts**. *Liu:* One Illumina 450 k dataset derives from the study Liu et al.[57], an EWAS for Rheumatoid Arthritis encompassing whole blood samples for 689 individuals (white Caucasian population). The raw data is available from GEO under accession number GSE42861. Batch normalized DNAm data was obtained from the authors and further adjusted for type-2 probe bias using BMIQ[53].

*Hannum:* Another Illumina 450k DNAm dataset derives from Hannum et al.[6], encompassing whole blood samples from 656 healthy individuals (426 white Caucasians and 230 Mexican Hispanic). The raw and normalized data is available from GEO under accession number GSE40279. Normalized DNAm data were further adjusted for type-2 probe bias using BMIQ[53].

*Buccal:* Another Illumina 450k dataset derived from 790 buccal swabs collected as part of the MRC1946 birth cohort NSHD study, and which was previously normalized and analyzed by us[9]. All 790 buccal swabs derive from women born in Britain in 1946 and were all collected at the same age (age = 53). This data is only available by submitting data requests to mrclha.swiftinfo@ucl.ac.uk; see full policy at http://www.nshd.mrc.ac.uk/data.aspx. Managed access is in place for this 74-year-old study to ensure that use of the data are within the bounds of consent given previously by participants, and to safeguard any potential threat to anonymity since the participants are all born in the same week.

*Tsaprouni:* This Illumina 450k DNAm dataset derives from Tsaprouni et al.[25], and consists of 464 whole blood samples from the CARDIOGENICS cohort, representing individuals of Caucasian ancestry, of which 226 were healthy and 238 had cardiovascular artery disease (CAD). Normalized data were downloaded from GEO under accession number GSE50660, and further adjusted for type-2 probe bias using BMIQ. CAD status information was not made available.

*ZhangHIV(450k/850k):* The Illumina 450k DNAm dataset derives from Zhang et al.[58], encompassing 608 blood samples from HIV patients. This population was predominantly white ($n = 522$) with 58 blacks and the 28 rest from other unspecified ethnicities. Normalized intensity values were downloaded from GEO under accession number GSE117859. The $\beta$ values (Illumina definition) were calculated using the provided unmethylated and methylated intensity values. Using the provided detection $P$ values, we first computed coverage per probe (fraction of samples with detection $P$ value < 0.05), removing low quality probes (coverage < 0.99) and subsequently computing coverage per sample over the good-quality probes, removing low quality samples (coverage < 0.99). The small remaining number of missing values were imputed using *impute.knn* (with $k = 5$) from the *impute* R-package[83]. The $\beta$ values were then corrected for type-2 probe bias using BMIQ. The $\beta$ values were then adjusted for type-2 probe bias using BMIQ. Beadchip effects were normalized using ComBat[54], as implemented in the sva R-package[84] The same Zhang et al.[58] study also used EPIC 850k beadarrays to profile an additional 529 whole blood samples, also from HIV patients (GSE117860). This population consisted of 426 whites, 48 blacks and 55 from other ethnic groups. The $\beta$ values (Illumina definition) were calculated using the provided unmethylated and methylated intensity values. We excluded samples and probes with more than 1% missing probes at detection $P$ value < 0.05. The small remaining number of missing values were imputed using *impute.knn* (with $k = 5$) from the *impute* R-package[83]. The $\beta$ values were then adjusted for type-2 probe bias using BMIQ. Beadchip effects were normalized using ComBat[54].

**Identification of smoking-associated DMCs and DMCTs**. In the TZH cohort we inferred both DMCs and cell-type specific DMCs, denoted DMCTs, whereas in the other cohorts we only inferred DMCTs. The inference of DMCTs proceeds via the CellDMC algorithm[40]. Briefly, CellDMC model runs the following linear model, which is run separately for each CpG $c$:

$$\overrightarrow{y_c} = \sum_{k=1}^{K} \mu_{ck} \overrightarrow{f_k} + \sum_{k=1}^{K} \beta_{ck}^{(I)} \overrightarrow{f_k} * \overrightarrow{z} + \overrightarrow{\varepsilon},$$

where $\overrightarrow{y_c}$ denotes the vector of DNAm values of CpG $c$ across all samples, $\overrightarrow{f_k}$ denotes the corresponding vector of cell-type fraction estimates for cell-type $k$ across all samples, $\overrightarrow{z}$ denotes the exposure of interest vector, $\mu_c$, $\mu_{ck}$, $\beta_c$, $\beta_{ck}^{(I)}$ are regression coefficients to be estimated, * denotes the interaction term, and where we assume $K$ cell-types and that errors are Gaussianly distributed with a variance that may depend on the specific CpG $c$. The regression coefficients $\beta_{ck}^{(I)}$ inform us as to whether there is a significant interaction between the exposure and the corresponding fraction for cell-type $k$. We note that if differential methylation associated with the phenotype occurs at a CpG $c$ and in cell-type $k$, that the observed differential methylation should be larger in samples with high fractions for that cell-type $k$ compared to samples with low content for cell-type $k$, and should be detectable via a statistically significant interaction term $\beta_{ck}^{(I)}$. We solve the above

model using least squares which provides estimates for the regression coefficients and their statistical significance via P values $P_{ck}^{(l)}$. The P values $P_{ck}^{(l)}$ for each cell-type $k$ are adjusted for multiple hypothesis testing using BH FDR estimation. For those CpGs with BH-adjusted P values less than a predefined significance threshold (i.e., typically BH FDR < 0.05), we call it a DMCT (differentially methylated cell-type) in the given cell-type. Finally, CpGs can be ranked within each cell-type according to the associated P value of significance. Finally, we note that additional covariates representing other biological (e.g., age, gender, and ethnicity) or technical factors (batch) can be included in the above model, as described by us previously[40].

Below we describe the specific implementations in each cohort:

*TZH:* Smoking-associated differentially methylated cytosines (DMCs) in our TZH whole blood cohort were identified by running multivariate linear models with DNAm as the dependent variable, smoking status (encoded as 0 for never-smokers, 1 for ex-smokers, and 2 for smokers at sample draw) as the exposure, and with age, sex, beadchip position, and blood cell type fractions (estimated using EpiDISH[56]) as covariates. Because the DNAm data matrix had already been adjusted for beadchip effects using ComBat, it was not necessary to adjust for this factor again. Regression t test P values were adjusted for multiple-testing using FDR estimates, as obtained using the q value Bioconductor package[85]. CellDMC was run with smoking status as exposure and with the same covariates, but at the resolution of 2 cell-types (i.e., $K=2$) representing generic lymphoid and myeloid cells. In the case of lymphocytes, we summed the estimated cell-type fractions of NK-cells, CD4 + T-cells, CD8+ T-cells and B-cells, whereas for the myeloid lineage, we added the cell-type fractions of monocytes, neutrophils and eosinophils.

*Liu:* Total myeloid and lymphoid fractions per sample were estimated by running EpiDISH[56] with our seven blood cell subtype reference DNAm matrix and then separately adding the fractions within the lymphoid and myeloid compartments. We then ran CellDMC by adjusting for rheumatoid arthritis status, age and gender, and with smoking status (encoded as never-smokers, ex-smokers, and current smokers) as the exposure of interest, and at a cell-type resolution level of two cell-types (myeloid and lymphoid).

*Hannum:* Total myeloid and lymphoid fractions per sample were estimated by running EpiDISH[56] with our seven blood cell subtype reference DNAm matrix and then separately adding the fractions within the lymphoid and myeloid compartments. We then ran CellDMC by adjusting for age and plate, with smoking status as the exposure of interest, and at a cell-type resolution level of two cell-types (myeloid and lymphoid). In the case of Hannum, adjusting for plate has the advantage that it also adjusts for center and ethnic group, as these were distributed in a plate-specific manner. Like in the TZH cohort, in Hannum we also observed a correlation between smoking-status and gender, and therefore decided against using sex as a covariate.

*Buccal:* In the case of the buccal swab cohort, we estimated total epithelial, total lymphoid and total myeloid fractions using HEpiDISH[35]. We then ran CellDMC at a resolution of these three cell-types with smoking status as the exposure of interest. In this cohort, no adjustment for age or gender is necessary because the buccal samples were all from women, and collected at the same age (53 years old). Beadchip and position effects were minor, and not deemed necessary to adjust for them, in line with our previous studies[9,18].

*Tsaprouni:* Total myeloid and lymphoid fractions per sample were estimated by running EpiDISH[56] with our seven blood cell subtype reference DNAm matrix and then separately adding the fractions within the lymphoid and myeloid compartments. We then ran CellDMC by adjusting for age and gender, with smoking status as the exposure of interest and at a cell-type resolution level of two cell-types (myeloid and lymphoid). We did not adjust for beadchip effects, because the dataset did not contain chip/batch information.

*ZhangHIV450k/850k:* Total myeloid and lymphoid fractions per sample were estimated by running EpiDISH[56] with our seven blood cell subtype reference DNAm matrix and then separately adding the fractions within the lymphoid and myeloid compartments. In these all male cohorts (Illumina 450 k and EPIC), we ran CellDMC by adjusting for age, with smoking status (encoded as nonsmokers and current smokers) as the exposure of interest, and at a cell-type resolution level of two cell-types (myeloid and lymphoid). Because the DNAm data matrix had already been adjusted for beadchip effects using ComBat, it was not necessary to adjust for this factor again.

**Meta-analysis**. Before running the meta-analysis, we clarify that polymorphic and cross-reactive probes were removed using the lists provided by Chen et al.[86]. CellDMC yields t-statistics of differential methylation in separate myeloid and lymphoid lineages and for each of the seven studies analyzed here. To obtain an aggregate meta-analysis P value for the common CpGs across all seven studies, we followed Efron's empirical Bayes procedure[59,60] to adjust the null-statistics in each study so as to ensure uniformity of these null-statistics across studies. Specifically, the t-statistic P values in each study were first transformed into corresponding z-statistics (i.e., quantiles of a normal distribution of mean 0 and standard deviation 1) taking into account the directionality of the t-statistics. For each study, we then used the *locfdr* R-package to estimate the mean $\mu$ and standard deviation $\sigma$ of the null z-statistics in each study. Modified z-statistics in each study were then defined by performing the z-score transformation $\rightarrow (z - \mu)/\sigma$. This guarantees that the null-statistics follow a N(0,1) distribution in each study, making the statistics more comparable between studies and thus avoiding study-specific biases due to

potential unaccounted confounding factors[60]. Next, for each of the common CpGs, an overall z-statistic is computed using Stouffer's method, i.e., by taking $z = \frac{1}{\sqrt{K}}\sum_{s=1}^{K} z_{(s)}$ where K is the number of studies ($K=7$) and where s labels the study. From these aggregate z-statistics we obtain corresponding meta-analysis P values, which we finally adjust for multiple testing using the well-known BH procedure to estimate the FDR. We also estimated the FDR by an independent method, where we randomized (i.e., permuted) the modified z-scores over all features in each study, subsequently deriving an empirical null for the aggregate z-statistics by performing the permutation operation a total of 1000 times and averaging the obtained aggregate z-statistics over the 1000 runs. An R-script, DoMetaEfron, implementing the above meta-analysis has been added to the EpiDISH Bioconductor package (http://www.bioconductor.org/packages/devel/EpiDISH).

**Simulation and sensitivity analysis**. We devised a simulation model in order to estimate the sensitivity to detect three separate categories of DMCTs: (i) DMCTs changing in both lymphoid and myeloid lineages (nonspecific DMCTs), (ii) DMCTs changing only in lymphoid cells (lymphoid-specific DMCTs), and (iii) DMCTs changing only in myeloid cells (myeloid-specific DMCTs). The DNAm profile of each sample was generated using Illumina 450 k DNAm profiles from Reynolds et al.[87], representing DNAm profiles for 214 CD4+ T-cell and 1202 monocyte samples. The raw idat files for this study are available from GEO under accession numbers GSE56581 and GSE56046, and were normalized with *minfi* and BMIQ as described by us previously[88], resulting in 482,077 common CpGs across all T-cell and monocyte samples. We considered two different sample size scenarios: $n = 200$ (100 cases and 100 controls) and $n = 600$ (300 cases and 300 controls). The first scenario guarantees statistical independence, since drawing 300 CD4+ T-cell profiles from only 214 samples can only be done with replacement, which would violate the statistical independence assumption. On the other hand, there is also the need to simulate for a more realistic sample size matched to real cohorts, as analyzed in this manuscript. Most of the cohorts analyzed in this study contain at least 600 samples. Each of the samples in our simulation were generated by taking an in silico mixture of two normalized DNAm profiles, one CD4+ T-cell sample and one monocyte sample, randomly selected from the population. The cell-type fractions were chosen randomly from realistic lymphoid myeloid fraction combinations, as estimated in our large TZH cohort. To define DMCTs, we first identified a total of 34,443 CpGs that were unmethylated (i.e., DNAm $\beta$ value < 0.1) across all 214 CD4+ T-cells and 1202 monocyte samples. We then randomly picked 1000 of these CpGs to be DMCTs. In cases, the DNAm values of DMCTs were drawn from a $\beta$-distribution with (a,b) parameters chosen to represent a range of different effect sizes, where the effect size is defined as

$$effect\ size = \Delta\mu / \sqrt{\frac{1}{2}(\sigma_1^2 + \sigma_2^2)}$$

where $\Delta\mu = |\mu_1 - \mu_2|$ is the absolute difference in average DNA methylation between case and control group, $\mu_1$ and $\mu_2$ are the mean DNA methylation levels in case and control group respectively, and where $\sigma_1$ and $\sigma_2$ are the standard deviation of DNA methylation levels in case and control groups. We chose eight parameter combinations (a,b) = (1,9) (0.5,2) (1,4) (2,8) (3,7) (4,6) (5,5) (7,3), where $\Delta\mu = |\mu_1 - \mu_2|$ ranged from 0.05 to >0.1, >0.2, >0.3, >0.4, >0.6. These eight parameter combinations lead to effect size estimates of around 1, 1.1, 1.4, 1.9, 2.7, 3.5, 4.4, and 6.8. We stress that these are effect sizes as measured in the individual cell-types, and do not represent the effect sizes observed in the mixtures. To justify that the range of effect sizes considered is realistic, we obtained estimates for the average effect size in whole blood from the TZH cohort. The mean effect size for the top 50 smoking-DMCs in whole blood was 1.7, for the top 100 it was 1.6, and for the top 1000 it was approximately 1. These effect sizes would also apply to individual lymphoid and myeloid cell compartments, provided the DMCTs occur in both lineages. However, if the DMCTs occur only in the myeloid linage, the above effect sizes would be inflated by a factor 10/7 (since the average myeloid fraction is around 70%), and so could range from 2.4 to 1.4. In the case of DMCTs only occurring in the lymphoid lineage, the above effect sizes would be inflated by a factor 10/3 (since the average myeloid fraction is around 30%), and so could range from 5.6 to 3.3. Thus, our choice of parameters, which leads to a range of effect sizes from approximately 1 to as high as 7, encompass realistic effect sizes at the resolution of myeloid and lymphoid cell-types. We ran a total of ten Monte-Carlo runs for each of the 8 parameter choices and for each of the 3 separate categories of DMCTs, recording their sensitivities, defined as the fraction of the 1000 DMCTs correctly predicted by CellDMC to be a DMCT in the corresponding cell-type. We also recorded the specificity, i.e., 1—false-positive rate (FPR), with the FPR defined as the fraction of false positives among all true non-DMCTs, as well as the precision or positive predictive value (PPV), defined as the fraction of true DMCTs among called DMCTs.

**Ethics**. The TZH cohort study was conducted with the official approval of the Ethics Committee of the Shanghai Institutes for Biological Sciences (ER-SIBS-261410). The Declaration of Helsinki Principles was followed and all participants provided written informed consent.

**Reporting summary**. Further information on experimental design is available in the Nature Research Reporting Summary linked to this paper.

## Data availability

All data analyzed in this paper are publicly available from GEO under accession numbers GSE42861 (Liu et al. dataset), GSE40279 (Hannum et al. dataset), GSE50660 (Tsaprouni et al. dataset), GSE117859 and GSE117860 (Zhang et al. dataset). The buccal swab DNAm data is only available by submitting data requests to mrclha.swiftinfo@ucl.ac.uk; see full policy at http://www.nshd.mrc.ac.uk/data.aspx. Managed access is in place for this 73-year-old study to ensure that use of the data are within the bounds of consent given previously by participants, and to safeguard any potential threat to anonymity since the participants are all born in the same week. The Illumina EPIC DNAm data for the TZH cohort can be viewed at NODE under accession number OEP000260, or directly at https://www.biosino.org/node/project/detail/OEP000260, and accessed by submitting a request for data-access. Data usage shall be in full compliance with the Regulations on Management of Human Genetic Resources in China. All other relevant data supporting the key findings of this study are available within the article and its Supplementary Information files or from the corresponding author upon reasonable request.

## Code availability

The code needed to run CellDMC and the meta-analysis are available in the EpiDISH Bioconductor R-package, which is freely available from http://bioconductor.org/packages/devel/EpiDISH. eFORGE was run with the webserver at https://eforge.altiusinstitute.org/. BMIQ was run using R-code, freely available from https://aeteschendorff-lab.github.io/software/BMIQ.

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

## Acknowledgements

A.E.T. was supported by NSFC (National Science Foundation of China) grants, grant nos. 31571359, 31771464, and 31970632. S. Wang and L.J. were supported by grants from Shanghai Municipal Science and Technology Major Project (2017SHZDZX01), Ministry of Science and Technology (2015FY111700), National Key Research and Development Project (2018YFC0910403) and the Strategic Priority Research Program of the Chinese Academy of Sciences (XDB38020400).

## Author contributions

Study was conceived by A.E.T. Statistical analyses were performed by C.Y. and A.E.T. with input from H.J., T.Z., S.C.Z., and S. Wu. Smoking information for Hannum cohort was provided by K.F. and G.W. Samples and DNAm data for the T.Z.H. cohort were generated and provided by S. Wang and J.L.

## Competing interests

The authors declare no competing interests.
