## [Peer Review File · Nature Communications]

Reviewers' comments:

Reviewer #1 (Remarks to the Author):

In this paper, the researchers perform a meta analysis for cell-type specific effects of smoking using their proposed method of cellDMC. cellDMC proposes that the interaction between cell type composition and the covariates are interpretable as the effect on the methylation of the unobserved cell type. They find a number of associations with myeloid cells, but only one in Lymphoid. More needs to be written about these types of methods and where they are susceptible to improper inference.

Major

* Treating smoking status as an ordinal categorical variable seems like a big assumption. Can you please provide justification for this.

* The citations seem like they are incorrect. I see a citation of 102 (line 470) but the references only go up to 69.

* How does the distribution of the Lymphoids and Myeloids compare between cohorts? It is mentioned that the Zhang et al [53] cohort is comprised of HIV patients. Do you have information on whether these patients are taking ART? I would imagine this would impact their cell type frequencies. This could cause differences in the cell type proportions between cohorts. Also, if there is sufficient heterogeneity within the cell type proportions this is likely to impact inference.

* Could the lack of consistency in the lymphoid associations be due to it being from the adaptive immune system which likely varies more between cohorts?

* Please add to the supplement more information about these different cohorts.

* Please provide descriptive statistics of the cohorts with information on the number of smokers, former, or never should be included.

* The methods could be better organized, potentially describing the studies, describing the pre-processing and then getting into the analysis.

* Is there the possibility of batch effects on the unobserved cell type methylations that somehow running combat would miss. What covariates did you pass to ComBat to control for?

* In the initial analysis there were 63,977 that had an evidence of $FDR < 0.05$, this then goes down to 170 once cellDMC is used. This seems like a large drop. Can the researchers provide some clarification on this.

* Line 195: I am unsure of the necessity of this section as once the researchers bring in the independent cohorts it focuses on the "gold standard CpG sites" or the results from Su et al. Is this included just due to it being on Epic? It seems like this section is redundant given the genome-wide meta analysis is not discussed on line 264.

* Is there any evidence of association of smoking with the cell type proportions? Would this cause

potential bias or instability in the results?

* Figure 1 B and C displays tests supposedly reporting the number with FDR <0.3 but the paper reports those numbers as the amount with FDR <0.1 (line 163). What is correct?

* Throughout the paper, it reads as if these methods are providing a causal interpretation which it is not. It provides evidence of an association correct?

* Multiple times throughout the paper the authors refer to smoking as the outcome of interest. It is the exposure of interest correct?

Minor

* Line 402: "vary very significantly". Please reword/clarify your language.

* Line 436: What was the k used for impute.knn.

*Line 118-120: This reads like a causal interpretation.

*Line 161: Please expand on how the qq-plot reveals evidence of a global association and not perhaps inflation.

* Language is confusing as some of the gold standard do not pass QC in the TZH study correct? Could use better clarification.

* Like 393: "Thus either ITGAL is a false positive finding of SU et al, or CellDMC ..". Could it be another option?

* Could be useful in the supplement to have information on what covariates were included in each model.

* Like 499: What was the distribution of smoking status by plate?

* Line 525: How were the null z-statistics defined?

Reviewer #2 (Remarks to the Author):

SUMMARY

The authors conduct a novel analysis on a DNA methylation data set from a cohort of 712 Chinese individuals, obtained via the Illumina MethylationEPIC array applied to whole blood. Using an algorithm previously developed by the authors, the analysis was able to separate effects of smoking on myeloid and lymphoid lineages, obtaining 7 lymphoid and 278 myeloid DMCs. In addition, the authors focused analysis on 60 smoking associated DMCs found by an independent group and replicated by other studies, replicating the finding of smoking related hypomethylation in a majority. The authors validated findings in three independent whole blood cohorts, an HIV cohort, a buccal swab study. They also conducted metaanalysis over all data sets. All of these analyses supported the claim that smoking principally affects DNA methylation in myeloid lineage cells at sites mapped to specific genes. Finally, the authors conduct a gene set enrichment analysis to demonstrate some global mechanisms that may be associated with the signature, demonstrating significant inflammatory effects.

MAJOR COMMENTS

Overall this is a well-written article that is easy to follow and quite convincing, using statistical techniques that are standard (except for a few minor details mentioned below). I applaud the authors for conducting novel analysis on a new data set while also replicating/verifying the work of others in detail. I also appreciate the buccal swab analysis which further demonstrates the limited impact of smoking on epithelial tissues (in contrast to myeloid lineage WBCs), as well as the analysis of data obtained from a population of individuals with substantially altered immune systems.

My one semi-major comment is that the Discussion should include some commentary on potential mechanisms. Certainly the eFORGE analysis demonstrates the inflammatory nature of the smoking-induced modifications (which certainly suggests an explanation for the many deleterious effects of smoking), but the story the authors are telling would be completed by a little more in-depth discussion of the molecular effects of smoking on the genes that are prominent within the proposed smoking signature. In other words, what are some hypotheses on the systemic processes that lead to hypomethylation of these specific genes? Some of the genes are well-studied (e.g. AHRR) but some of them are less so. Such a discussion would help convey the systemic meaning of these changes and may provide clues for potential replication/verification using methods orthogonal to DNA methylation.

MINOR COMMENTS

1. Line 162: Why was $FDR < 0.1$ used? Also, Figure 1b refers to a threshold of 0.3 instead of 0.1, so there seems to be a discrepancy between the text and the figure. In general, not all of the FDR thresholds (which vary across the paper) are given a clear rationale.
2. Figure 4a: the x-axis labels are confusing. Do they refer to specific columns in the heat map? If so, maybe tick marks would help clarify. Is the P-value row an annotation track or a legend? I suspect the former, but if so, were the columns sorted by P-value or using hierarchical clustering? More details are needed.
3. Lines 433-436: some additional details would be helpful. How was total intensity used to obtain p-values? Which list of SNPs? If these are all standard settings, there should be language that says so. Also, why was detection $P < 0.01$ as a threshold here but $P < 0.05$ for other data sets?
4. Line 522: I think "Methods" should probably be deleted since it seems like a circular reference here.

Reviewer #3 (Remarks to the Author):

The authors showed that smoking-related DNA methylation changes reported in whole blood are driven by corresponding alterations in myeloid cells. The authors applied a novel cell-type deconvolution algorithm (CellDMC) that identified the differentially methylated cell-types in seven large independent cohorts. The authors showed that results were consistent across Chinese and Caucasian populations. They also performed a lineage-specific meta-analysis over the seven studies that revealed additional novel insights, including the identification of a novel myeloid-specific smoking-associated hypermethylation signature enriched for DNase Hypersensitive Sites in acute myeloid leukemia.

This manuscript is well written and highly relevant to the field. This manuscript may help guide the design of future smoking studies.

Reviewer #4 (Remarks to the Author):

Previous studies have established that highly reproducible smoking-associated DNA methylation differences are present in white blood cells. This study makes a valuable contribution by examining cell-type specific associations with smoking. My comments are listed below.

The abstract states "A meta-analysis further reveals a novel myeloid-specific smoking-associated hypermethylation signature enriched for DNase Hypersensitive Sites in acute myeloid leukemia." Although I recognize that the myeloid-specific finding and link to myeloid leukemia is novel, I wonder how many of the methylation sites identified by the lineage-specific meta-analysis have been picked up before in previous large studies of smoking, in particular the meta-analysis of smoking of 15907 individuals by Joehanes et al 2016 (PMID 27651444)?

Introduction

"Indeed, we recently confirmed this for the case of buccal swabs, a tissue that consists of approximately 50% squamous epithelial and 50% immune-cell infiltrates"

I believe this sentence could be more nuanced, as this proportion appears to vary across datasets, with some datasets showing a much higher proportion of epithelial cells, see:

Eipel M, Mayer F, Arent T, Ferreira MRP, Birkhofer C, Gerstenmaier U, et al. Epigenetic age predictions based on buccal swabs are more precise in combination with cell type-specific DNA methylation signatures. *Aging (Albany NY)*. 2016;8:1034–48. 23.

Theda C, Hwang SH, Czajko A, Loke YJ, Leong P, Craig JM. Quantitation of the cellular content of saliva and buccal swab samples. *Sci Rep. Springer US*; 2018;8:6944.

Van Dongen, J, et al. "Genome-wide analysis of DNA methylation in buccal cells: a study of monozygotic twins and mQTLs." *Epigenetics & chromatin* 11.1 (2018): 1-14.

Page 4: "Of the 62 gold-standard smoking-associated CpGs derived from a previous meta-analysis over white Caucasian cohorts[8]"

* I believe this [Gao et al, reference 8] is not a meta-analysis but a review?

* Also, I wonder why the authors have used the list of 62 'gold-standard' smoking-CpGs from the review by Gao et al to benchmark their findings, while they could have used the > 2000 CpGs from the more recent and largest meta-analysis to date by Joehanes et al 2016 (PMID 27651444)?

Page 4. The authors conclude that "smoking affects DNAm patterns largely independently of genotype/ethnicity" based on their finding that the gold-standard CpGs are also significant in this Asian population with the same direction of effect. I believe that a more comprehensive analysis would be required to support such a statement. I wonder if the effect size are also exactly the same across populations? I realize that a comparison of effect sizes between populations might not only be affected by genotype/ethnicity but also by differences in smoking intensity between populations. The authors might want to consider nuancing this statement.

How do the authors explain the large discrepancy between the number of CpGs identified in the ordinary EWAS analysis in the TZH cohort (63977) versus the cell lineage-specific analysis (170 myeloid-specific and no lymphoid-specific). Could they comment on this in the discussion? Does this reflect the lower power of the CellDMC interaction model or does it mean that the majority of smoking-associated methylation signal is in fact NOT lineage-specific? If a methylation difference is present in both the lymphoid and myeloid lineage, how does this present in CellDMC?

Page 6 "Of the 60 hypomethylated gsSMK-CpGs, 53 passed QC". It is unclear to me why this is mentioned here, since the gsSMK-CpGs were also already discussed on page 4 (fig1A).

Page 6 "and a substantial fraction of these exhibited hypomethylation in the myeloid..". could the authors be more specific (e.g. give the exact percentage in this sentence)

Page 6 "A similar trend was observed in the lymphoid cells, albeit not as strong as in the myeloid compartment". At this point, it is unclear what is meant with 'similar trend' (although this does become clear to me later in this paragraph). For clarity, could similar trend be replaced by something like 'showed the same direction of effect'?

Maybe I missed it, but could the total sample size of the meta-analysis of 7 cohorts be presented somewhere?

Page 13 "approximately 70 CpGs" sounds vague. Could the exact number be stated?

Page 14 (discussion). The authors mention that it will be interesting to apply CellDMC to thousands of samples merged together in one dataset. If this would be possible, I wonder why they chose to apply a meta-analysis approach in the current paper?

Methods, page 15 "common SNPs were removed using minfi". Which reference population did the authors use (is the common SNP filter in minfi suited for Asian populations?)?

Software availability. The CellDMC tool is a highly valuable tool for the research community and it is great that this software package is freely available. Will the authors also make their script to perform meta-analysis of cell-type specific EWAS test statistics available? This would be highly valuable.

Reviewers' comments:

Reviewer #1 (Remarks to the Author):

General: In this paper, the researchers perform a meta analysis for cell-type specific effects of smoking using their proposed method of cellDMC. cellDMC proposes that the interaction between cell type composition and the covariates are interpretable as the effect on the methylation of the unobserved cell type. They find a number of associations with myeloid cells, but only one in Lymphoid. More needs to be written about these types of methods and where they are susceptible to improper inference.

Response: We sincerely thank the reviewer for taking time to evaluate our manuscript and for the feedback provided, which has helped us improve this manuscript. The reviewer is right that we should have written more about these types of methods and how susceptible they are to proper inference. In response to this, we have briefly expanded on this in the Introduction. We have also added a new subsection and a new Figure-5 displaying the results of a power calculation derived from a simulation analysis, which we think helps to interpret our findings. Reason why we however don't wish to delve too much into this particular aspect, is simply because this manuscript is not a methods paper where we compare algorithms. We already have a separate manuscript dealing with that particular aspect. In Discussion, we also alert the reader to some of the potential limitations of a method like CellDMC, and provide in-depth discussion of how the results in this manuscript should be interpreted.

Major points:

** Treating smoking status as an ordinal categorical variable seems like a big assumption. Can you please provide justification for this.*

Response: The reviewer is right that we have encoded smoking exposure as ordinal with 0 indicating never-smoker, 1 ex-smoker and 2 current smoker. The obvious alternative measure to use, if available, is smoking-pack-years (SPY) which would be effectively a continuous variable. Indeed, in some of our previous studies (e.g. Teschendorff AE et al JAMA Oncology 2015) we used SPY. It is worth pointing out here however that in our experience results and conclusions are largely independent of whether we had used SPY or an ordinal smoking variable. The reason why in this work we chose to encode smoking as an ordinal variable is simply because here we are conducting a meta-analysis over many studies, and SPY information was not available for several of these cohorts. In addition, in those cohorts where SPY is available, there are significant numbers of smokers and ex-smokers with no SPY entry, which may lead to a loss of power. Finally, we should point out that SPY itself, even if available for all cohorts and samples, is also not ideal because for ex-smokers the time from quitting to sample draw has also been shown to be important. The complexity of how best to encode smoking exposure can't be underestimated and we believe that this is a very interesting question to explore in future and in cohorts that have full smoking history information. In response to the reviewer's point, we have

added a sentence to the main text explaining why smoking was treated as ordinal.

* *The citations seem like they are incorrect. I see a citation of 102 (line 470) but the references only go up to 69.*

Response: We thank the reviewer for pointing this out to us. This was indeed a typo, and we have corrected that.

* *How does the distribution of the Lymphoids and Myeloids compare between cohorts? It is mentioned that the Zhang et al [53] cohort is comprised of HIV patients. Do you have information on whether these patients are taking ART? I would imagine this would impact their cell type frequencies. This could cause differences in the cell type proportions between cohorts. Also, if there is sufficient heterogeneity within the cell type proportions this is likely to impact inference.*

Response: In response to this, we have compared the lymphoid to myeloid proportions across the 6 whole blood cohorts, which has not revealed major differences between them. The HIV cohorts (ZhangHIV850k&450k) do show a marginally lower myeloid and higher lymphoid count compared to the other cohorts but difference is not big. For convenience, we display this figure (now new Suppl.Fig.2) below:

With regards to the HIV cohorts, these HIV patients were given ART, but not all patient complied or adhered to it. We have checked whether cell-type fractions show any differences with respect to ART adherence or not, and we did not find any. This data is now shown in new SuppFigs.3-4. For this reason we also never adjusted for ART in our analysis, since this factor did not impact on DNAm patterns.

* *Could the lack of consistency in the lymphoid associations be due to it being from the adaptive immune system which likely varies more between cohorts?*

Response: We thank the reviewer for raising out this possibility. With the exception of the Zhang et al HIV cohort, all other cohorts are from generally healthy individuals, so we are not sure which specific differences between cohorts could explain the lack of consistency in the lymphoid associations, specially given that these are fairly large cohorts of individuals. That lymphoid cells are part of the adaptive immune response is however an important point as lymphoid progenitor cells from the bone marrow migrate to the thymus where they undergo maturation. We can speculate that the maturation process in the thymus may erase smoking-associated marks acquired by the progenitors in the bone-marrow. In response to the reviewer's excellent point, we

have added one sentence to the Discussion to highlight this possibility.

* Please add to the supplement more information about these different cohorts. Please provide descriptive statistics of the cohorts with information on the number of smokers, former, or never should be included.

Response: We agree that we should have provided more details. These can now be found in a new Supplementary Table that can be found in Supplementary Data (Suppl.Data.1)., which for convenience we display below:

SuppTableS1: Participant Characteristics												
	TZH (n=688)			Liu (n=686)			Hannum (n=588)			Tsaprouni (n=463)		
	n	Male (%)	Age	n	Male (%)	Age	n	Male (%)	Age	n	Male (%)	Age
Never-smokers	453 (66%)	27%	54+/-10	193 (28%)	23%	51+/-14	384 (65%)	40%	63+/-15	179 (39%)	68%	55+/-7
Ex-smokers	62 (9%)	97%	60+/-8	294 (43%)	34%	52+/-11	178 (30%)	57%	69+/-13	263 (57%)	73%	55+/-7
Smokers	173 (25%)	98%	55+/-10	199 (29%)	26%	53+/-10	26 (5%)	62%	54+/-12	21 (4%)	67%	54+/-6
	Zhang (IV450k) (n=608)			Zhang (IV850k) (n=529)			Buccal (n=790)					
	n	Male (%)	Age	n	Male (%)	Age	n	Male (%)	Age			
Never-smokers	247 (41%)	100%	50+/-9	220 (42%)	100%	48+/-9	258 (33%)	0%	53			
Ex-smokers	0 (0%)	NA	NA	0 (0%)	NA	NA	365 (46%)	0%	53			
Smokers	361 (59%)	100%	49+/-7	309 (58%)	100%	50+/-6	173 (22%)	0%	53			

* The methods could be better organized, potentially describing the studies, describing the pre-processing and then getting into the analysis.

Response: We appreciate the reviewer's point. In response to this, we have reorganized the Methods section in order to better describe the individual studies and their preprocessing.

* Is there the possibility of batch effects on the unobserved cell type methylations that somehow running combat would miss. What covariates did you pass to ComBat to control for?

Response: We thank the reviewer for raising this excellent question. Usually, by batch effects we mean technical factors such as chip/slide, chip position, plate, laboratory etc. It is very reasonable to assume that these technical factors would not affect cell-types differently, since their effects are imparted on the whole blood sample, and therefore it would affect all underlying cell-types equally. ComBat is an excellent Bayesian approach for correcting batch effects and it was designed primarily for categorical factors where the number of samples in each batch is small and where a Bayesian approach is desirable. Therefore, our policy is to apply ComBat to adjust for beadchip, assuming that beadchip effects are present, because for Illumina beadchips, each chip can only accommodate 12 or 8 samples depending on the methylation beadchip version. For other batch effects where the number of samples is considerably larger (for the cohorts considered in this paper, these would include plate, ethnicity and beadchip position) we tend to adjust for these by including a fixed effect covariate in the model. Overall, given the excellent consistency we observe in terms of the smoking-associated CpGs we find in whole blood, and separately also in the myeloid lineage, we feel that the specific batch effect correction done in each study has worked well. In response to the reviewer's point we have improved the description of the batch correction procedure in each study.

**In the initial analysis there were 63,977 that had an evidence of FDR <0.05, this then goes down to 170 once cellDMC is used. This seems like a large drop. Can the researchers provide some clarification on this.*

Response: We thank the reviewer for raising this excellent question. The answer comes in two parts. First, we acknowledge that the 63977 DMCs at FDR<0.05 as inferred in the TZH cohort may be an overestimate. Even if we were to use a Bonferroni threshold, we would still have 5308 smoking-DMCs in the TZH cohort. We note that although the recent large meta-analysis study of Joehanes et al (now cited and discussed in the revised manuscript) discovered at Bonferroni significance on the order of 2600 smoking-DMCs, we feel that this may also be an overestimate. Indeed, we note that over 50% of the presumed 2600 gold-standard smoking-DMCs from Joehanes et al did not validate in the TZH cohort (this is now mentioned in the main text and related data is shown in a new Fig.1b), so we suspect that the true number of smoking-associated DMCs is lower, and probably not more than 1000. We note that inflation is an ubiquitous feature of these studies, as it is extremely difficult to avoid the effect of hidden unknown confounders. An unknown study-specific confounder could easily inflate statistical significance estimates, even in the context of a meta-analysis study. We further note that the study by Gao & Brenner (Clin Epigenet 2015) who performed a review of 12 whole blood smoking-EWAS, only arrived at a relatively small number of gold-standard smoking-DMCs: approximately 151 smoking-DMCs were observed in at least 2 studies, and 62 in at least 3 of the 12 studies. Thus, although it is clear that individual studies may report on the order of thousands of DMCs, when we assess the consistency across independent studies we arrive at a much smaller number. We further note that almost all of these 62 CpGs were significant in the TZH cohort (as shown in Fig.1a), which if compared with the less than 50% that validated from the larger list of ~2600 smoking-DMCs from Joehanes et al, suggests that indeed the number of true smoking-DMCs (as assessed in whole blood) is on the order of hundreds. In the TZH cohort itself, one potential confounder which could explain the inflation, but which we are unable to adjust for, is sex. Indeed, because in this Chinese cohort over 95% of smokers are males (with the proportion of males in non-smokers being only 27%) adjusting for sex would deflate and effectively remove the well-known associations with smoking (e.g. many of the AHRR hits would disappear if we were to adjust for sex, which would contradict a large body of literature). While we could have run the analysis in the TZH cohort using only males, this would however render the study less powerful to detect associations with smoking, most of which are probably also sex-independent. Indeed, as far as the reported gold-standard lists of smoking-associated DMCs is concerned, there is not much evidence that these are sex-specific.

Now to the second part of our answer: even if the true number of smoking-DMCs in whole blood is around 1000, when running CellDMC to find cell-type specific DMCs (i.e. DMCTs), we still observe a substantial drop in the number of hits, as correctly pointed out by the reviewer. Addressing this issue is complex because for most loci it is not known in which cell-types they are changing. While we provided sensitivity

measures for a wide range of different scenarios in our CellDMC paper (see Zheng SC et al Nat Methods 2018), we have now conducted a more focused simulation in order to explain the above phenomenon. Smoking-DMCTs could be of 3 types: (i) they change in both myeloid and lymphoid lineages (non-specific DMCTs), (ii) they only change in the myeloid lineage (myeloid-specific DMCTs), (iii) they only change in the lymphoid lineage (lymphoid-specific DMCTs). We have run simulations for the 3 different scenarios for realistic sample sizes, cell-type fractions (myeloid fraction is higher than lymphoid fraction) and effect sizes, and the results of these simulations (shown in a new Figure-5) indicate that we have adequate power to detect lymphoid-specific or myeloid-specific DMCTs if these exist. However, the simulations also demonstrate that the power to detect DMCTs within the lymphoid lineage is compromised if these same loci are also changing in the myeloid-lineage. This happens because the myeloid fraction is higher.

To summarize, we believe that the large drop in hits when going from DMCs to DMCTs owes mainly to inflation at the level of DMCs. Although CellDMC appears to have adequate power to detect lineage-specific DMCTs, power estimates depend on the effect sizes within individual cell-types, which are largely unknown. Running CellDMC on datasets profiling thousands of samples together may help shed further light on this question. In response to the reviewer's point, we have a new subsection in Results discussing the results of our simulation, and a new Fig.5 displaying the power calculation results of this simulation, and have also expanded Discussion to better interpret our results in the light of the simulation analysis.

** Line 195: I am unsure of the necessity of this section as once the researchers bring in the independent cohorts it focuses on the "gold standard CpG sites" or the results from Su et al. Is this included just due to it being on Epic? It seems like this section is redundant given the genome-wide meta analysis is not discussed on line 264.*

Response: The reviewer is right that the way we have structured the results is largely motivated by the fact that we started out with a Chinese cohort profiled on EPIC arrays, whereas the other cohorts are non-Chinese and 450k. Before running a meta-analysis, we wanted to check if the results obtained in the EPIC set would validate in other cohorts, and if they were consistent with those in Su et al. The focus on gold-standard CpGs is simply because these did pop out from the analysis in the Chinese cohort. We think that it is very important to have this separate section dedicated to the gold-standard CpGs and also the CpGs assessed in Su et al, in order to emphasize that (i) the results obtained with CellDMC in each of the separate cohorts are highly consistent with the results of Su et al (Fig.3), and (ii) that the gold-standard CpGs are hypomethylated predominantly in the myeloid lineage. This then provides the justification to perform a meta-analysis. If we were to perform a meta-analysis first we worry that some of these key messages would be diluted out.

** Is there any evidence of association of smoking with the cell type proportions? Would this cause potential bias or instability in the results?*

Response: We thank the reviewer for raising this question. We did not observe any association between smoking status and the lymphoid & myeloid fractions. Below we display the figure for the TZh, Liu and Hannum cohorts (similar pattern is seen for the other 3 blood cohorts):

In any case, let us also point out to the reviewer that the linear model implemented in CellDMC contains terms that adjust for the cell-type fractions. Hence, the interaction between exposure and cell-type fraction is being assessed after adjustment of the DNAm data for the cell-type fraction itself. Thus, assuming there would be an association between smoking and cell-type fraction, this would not cause any problem to the inference. Indeed, we did address this issue already in our previous publication (see Zheng SC et al Nat Methods 2018) where CellDMC was presented. In response to the reviewer's point, we have now added more details about CellDMC to the Methods section.

* Figure 1 B and C displays tests supposedly reporting the number with FDR <0.3 but the paper reports those numbers as the amount with FDR <0.1 (line 163). What is correct?

Response: We thank the reviewer for pointing this error out to us. This was indeed a typo in old Fig.1b, as the correct threshold used is FDR<0.1 . We have now corrected this in what is now new Fig.1c.

* Throughout the paper, it reads as if these methods are providing a causal interpretation which it is not. It provides evidence of an association correct?

Response: We sincerely apologize if we have given this impression, but are also unsure in what way we had implied causality. This manuscript only considers the effect of an exposure (smoking) on DNA methylation patterns. The DMCTs we find are purely associative. In response to the reviewer's point, we have revised the language whenever we felt there could be confusion.

** Multiple times throughout the paper the authors refer to smoking as the outcome of interest. It is the exposure of interest correct?*

Response: The reviewer is absolutely right and we apologize for being imprecise. We have corrected this in the revised version.

Minor points:

** Line 402: "vary very significantly". Please reword/clarify your language.*

Response: We have now modified this sentence.

** Line 436: What was the k used for impute.knn.*

Response: We apologize for this omission. We used k=5, and have now stated this in the revised version.

**Line 118-120: This reads like a causal interpretation.*

Response: We apologize for the inaccurate language. We have now rephrased this.

**Line 161: Please expand on how the qq-plot reveals evidence of a global association and not perhaps inflation.*

Response: The statistics shown in a qq-plot relate the observed P-values to the expected ones under the null hypothesis of no global association. This plot is therefore deeply intertwined with the FDR estimates, which we added to the figures precisely in order to demonstrate that there appears to be a global association. Inflation is unlikely to have happened here since many of the top hits in the myeloid lineage are well-known gsSMK-CpGs.

** Language is confusing as some of the gold standard do not pass QC in the TZH study correct? Could use better clarification.*

Response: The reviewer is right that because the gold-standard CpGs were derived from a meta-analysis of 450k datasets, that not all of these are present on the EPIC array. We have verified that of the 62 gold-standard CpGs, 55 are on the EPIC array, and 54 passed QC. Of the 60 hypomethylated gold-standard ones, 54 are on the EPIC array and 53 passed QC. In other words, the QC step only removed 1 gold-standard CpG. We have now clarified this in the main text.

** Like 393: "Thus either ITGAL is a false positive finding of SU et al, or CellDMC ..". Could it be another option?*

Response: Indeed, we agree that there could be another option, and we did explicitly say that an alternative explanation is that ITGAL, because it only appears to have change in B-cells, that it would be much harder for CellDMC to detect it, since B-cells

only represent a smaller fraction of all lymphoid cells (in contrast to GPR15, which CellDMC does detect and which according to Su et al changes in both T-cells and B-cells).

* *Could be useful in the supplement to have information on what covariates were included in each model.*

Response: In response to this, we have now improved the presentation of the implementation of CellDMC in each cohort, clearly specifying which covariates were used. The improved description of the cohorts, their normalization and implementation of CellDMC can be found in the Methods section of the manuscript.

* *Like 499: What was the distribution of smoking status by plate?*

Response: Plate information is not available for the Liu et al cohort. In any case, with the Liu et al dataset, we obtained the normalized DNAm dataset (i.e. normalized for beadchip and plate) from the author himself.

* *Line 525: How were the null z-statistics defined?*

Response: We apologize for not explaining this. From running CellDMC in each cohort, we have a t-statistic and P-value for each CpG in each cell-type. This P-value can now be transformed into a statistic of a standard normal distribution of mean 0 and std.dev=1, which is what we mean by “z-statistic”. We have now clarified this in the methods section.

Reviewer #2 (Remarks to the Author):

General: The authors conduct a novel analysis on a DNA methylation data set from a cohort of 712 Chinese individuals, obtained via the Illumina MethylationEPIC array applied to whole blood. Using an algorithm previously developed by the authors, the analysis was able to separate effects of smoking on myeloid and lymphoid lineages, obtaining 7 lymphoid and 278 myeloid DMCs. In addition, the authors focused analysis on 60 smoking associated DMCs found by an independent group and replicated by other studies, replicating the finding of smoking related hypomethylation in a majority. The authors validated findings in three independent whole blood cohorts, an HIV cohort, a buccal swab study. They also conducted metaanalysis over all data sets. All of these analyses supported the claim that smoking principally affects DNA methylation in myeloid lineage cells at sites mapped to specific genes. Finally, the authors conduct a gene set enrichment analysis to demonstrate some global mechanisms that may be associated with the signature, demonstrating significant

inflammatory effects.

Response: We sincerely thank the reviewer for evaluating our manuscript and for the generally positive feedback.

Major points:

Comment: Overall this is a well-written article that is easy to follow and quite convincing, using statistical techniques that are standard (except for a few minor details mentioned below). I applaud the authors for conducting novel analysis on a new data set while also replicating/verifying the work of others in detail. I also appreciate the buccal swab analysis which further demonstrates the limited impact of smoking on epithelial tissues (in contrast to myeloid lineage WBCs), as well as the analysis of data obtained from a population of individuals with substantially altered immune systems.

Response: We thank the reviewer for the positive and encouraging feedback.

Comment: My one semi-major comment is that the Discussion should include some commentary on potential mechanisms. Certainly the eFORGE analysis demonstrates the inflammatory nature of the smoking-induced modifications (which certainly suggests an explanation for the many deleterious effects of smoking), but the story the authors are telling would be completed by a little more in-depth discussion of the molecular effects of smoking on the genes that are prominent within the proposed smoking signature. In other words, what are some hypotheses on the systemic processes that lead to hypomethylation of these specific genes? Some of the genes are well-studied (e.g. AHR) but some of them are less so. Such a discussion would help convey the systemic meaning of these changes and may provide clues for potential replication/verification using methods orthogonal to DNA methylation.

Response: We agree with the reviewer that we should have explored some of the genes in more detail and to also provide some discussion on their possible significance. In response to this, we have now done so, both in Results as well as in Discussion, and in relation to both the myeloid specific hypomethylation and hypermethylation signatures.

Minor points:

1. Line 162: Why was $FDR < 0.1$ used? Also, Figure 1b refers to a threshold of 0.3 instead of 0.1, so there seems to be a discrepancy between the text and the figure. In general, not all of the FDR thresholds (which vary across the paper) are given a clear rationale.

Response: The reviewer is right and we have corrected the error in the Figure. It should have read $FDR < 0.1$. We chose $FDR < 0.1$, because at this threshold there were hits in the lymphoid lineage, i.e. at $FDR < 0.05$, we got no hits in the lymphoid lineage, as mentioned in the main text. There is nothing magical about a 0.05 or 0.07 or 0.1

threshold. A FDR<0.1 threshold means that among the 7 hits, 10% may be false positives assuming no other confounders are present.

2. Figure 4a: the x-axis labels are confusing. Do they refer to specific columns in the heat map? If so, maybe tick marks would help clarify. Is the P-value row an annotation track or a legend? I suspect the former, but if so, were the columns sorted by P-value or using hierarchical clustering? More details are needed.

Response: We apologize for the confusion. Each column in the heatmap and in the P-value annotation bar represents one of the 107 CpGs. The CpGs (i.e. DMCTs) have been ordered according to the meta-analysis P-value. We had annotated some of these CpGs with the corresponding gene they map to. We have now placed the gene names closer to the annotation track and have also added the word "CpGs" to the top.

3. Lines 433-436: some additional details would be helpful. How was total intensity used to obtain p-values? Which list of SNPs? If these are all standard settings, there should be language that says so. Also, why was detection $P<0.01$ as a threshold here but $P<0.05$ for other data sets?

Response: The computation of detection P-values is a fairly standard procedure and is implemented and described in the *minfi* BioC R-package. Briefly, the total intensity U+M is compared to that of the background distribution as given by negative control probes. Probes with SNPs (i.e. mapping to the interrogated CpG or the Single-Base-Extension) were removed using the minfi package which uses common SNP tables from UCSC and the EPIC Illumina annotation file. As far as the P-value threshold is concerned, for the EPIC array we used 0.01 following the recommendation in the minfi package. However, for the 450k sets we have in the past used $P<0.05$, and we decided to stick to the same threshold here. We note that using a more stringent threshold in the EPIC set did not compromise coverage. We have now clarified some of these points in the methods section.

4. Line 522: I think "Methods" should probably be deleted since it seems like a circular reference here.

Response: We thank the reviewer for pointing out this typo, which we have now corrected.

Reviewer #3 (Remarks to the Author):

Comment: The authors showed that smoking-related DNA methylation changes reported in whole blood are driven by corresponding alterations in myeloid cells. The authors applied a novel cell-type deconvolution algorithm (CellDMC) that identified the differentially methylated cell-types in seven large independent cohorts. The authors showed that results were consistent across Chinese and Caucasian populations. They also performed a lineage-specific meta-analysis over the seven studies that revealed additional novel insights, including the identification of a novel

myeloid-specific smoking-associated hypermethylation signature enriched for DNase Hypersensitive Sites in acute myeloid leukemia. This manuscript is well written and highly relevant to the field. This manuscript may help guide the design of future smoking studies.

Response: We would like to thank the reviewer for taking time to evaluate our manuscript, for appreciating the importance of this study and for the positive feedback provided.

Reviewer #4 (Remarks to the Author):

General Comment: Previous studies have established that highly reproducible smoking-associated DNA methylation differences are present in white blood cells. This study makes a valuable contribution by examining cell-type specific associations with smoking.

Response: We would like to thank the reviewer for taking time to evaluate our manuscript and recognizing the valuable contribution it makes.

Comment: The abstract states “A meta-analysis further reveals a novel myeloid-specific smoking-associated hypermethylation signature enriched for DNase Hypersensitive Sites in acute myeloid leukemia.” Although I recognize that the myeloid-specific finding and link to myeloid leukemia is novel, I wonder how many of the methylation sites identified by the lineage-specific meta-analysis have been picked up before in previous large studies of smoking, in particular the meta-analysis of smoking of 15907 individuals by Joehanes et al 2016 (PMID 27651444)?

Response: We are very grateful to the reviewer for pointing out to us this excellent meta-analysis paper, and we sincerely apologize for having missed this important work. The question raised by the reviewer is also excellent. In response to this, we have now added a new Suppl.Table (Suppl.Data.4) that display the fractions of CpGs identified in our meta-analysis that are also found in the list from Joehanes. This table also demonstrates the strong consistency in terms of the direction of effect: for instance, of the 448 myeloid-DMCTs (FDR<0.3) from our meta-analysis, 363 (i.e 81%) were found in the list of Joehanes (FDR<0.05), with these 363 exhibiting 100% agreement in terms of directionality (i.e 195 hypomethylated in both meta-analyses and 168 hypermethylated in both). For the 5 lymphoid-DMCTs, 4 were found in the list by Joehanes, once again with 100% agreement in terms of directionality (all hypomethylated in both). For convenience we display the SuppData4 table below:

	Joehanes HypoM	Joehanes HyperM	Joehanes
Myeloid-DMCTs (FDR<0.3) n=448	195	168	363 (81%)
# Studies HypoM = 7	140	0	
# Studies HypoM = 6	51	0	

# Studies HypoM = 5	4	0	
# Studies HyperM = 5	0	19	
# Studies HyperM = 6	0	89	
# Studies HyperM = 7	0	60	
	Joehanes HypoM	Joehanes HyperM	Joehanes
Lymphoid-DMCTs (FDR<0.3) n=5	4	0	4 (80%)
# Studies HypoM = 7	2	0	
# Studies HypoM = 6	2	0	
# Studies HypoM = 5	0	0	
# Studies HyperM = 5	0	0	
# Studies HyperM = 6	0	0	
# Studies HyperM = 7	0	0	

Comment: Introduction “Indeed, we recently confirmed this for the case of buccal swabs, a tissue that consists of approximately 50% squamous epithelial and 50% immune-cell infiltrates” I believe this sentence could be more nuanced, as this proportion appears to vary across datasets, with some datasets showing a much higher proportion of epithelial cells, see: Eipel M, Mayer F, Arent T, Ferreira MRP, Birkhofer C, Gerstenmaier U, et al. Epigenetic age predictions based on buccal swabs are more precise in combination with cell type-specific DNA methylation signatures. *Aging (Albany NY)*. 2016;8:1034–48. 23. Theda C, Hwang SH, Czajko A, Loke YJ, Leong P, Craig JM. Quantitation of the cellular content of saliva and buccal swab samples. *Sci Rep. Springer US*; 2018;8:6944. Van Dongen, J, et al. "Genome-wide analysis of DNA methylation in buccal cells: a study of monozygotic twins and mQTLs." *Epigenetics & chromatin* 11.1 (2018): 1-14.

Response: The reviewer is absolutely right that there is substantial variation in the estimated epithelial fraction of buccal swabs, as reported in the literature. In response to this, we have accordingly modified the sentence in the introduction citing the above 3 references.

Comment: Page 4: “Of the 62 gold-standard smoking-associated CpGs derived from a previous meta-analysis over white Caucasian cohorts[8]” * I believe this [Gao et al, reference 8] is not a meta-analysis but a review? * Also, I wonder why the authors have used the list of 62 ‘gold-standard’ smoking-CpGs from the review by Gao et al to benchmark their findings, while they could have used the > 2000 CpGs from the more recent and largest meta-analysis to date by Joehanes et al 2016 (PMID 27651444)?

Response: The reviewer is absolutely right that the list of 62 CpGs derives from a review, not a meta-analysis. We have now corrected this. As to why we had focused on this list and not on the list from Joehanes, this is mainly down to the fact that we

were not fully aware of the study of Joehanes et al. We recognize that this is a very important study, and so in response to the reviewer's point we have now included the corresponding list of 2622 smoking-associated CpGs in our benchmarking: (1) Figure-1 has been extended to include two additional panels displaying results for these 2622 CpGs in the TZH cohort. (2) We have a new Suppl.Table (Suppl.Data.4) (see earlier response) which evaluates how many of our myeloid and lymphoid specific DMCTs from the meta-analysis are found in the meta-analysis by Joehanes.

Comment: Page 4. The authors conclude that "smoking affects DNAm patterns largely independently of genotype/ethnicity" based on their finding that the gold-standard CpGs are also significant in this Asian population with the same direction of effect. I believe that a more comprehensive analysis would be required to support such a statement. I wonder if the effect size are also exactly the same across populations? I realize that a comparison of effect sizes between populations might not only be affected by genotype/ethnicity but also by differences in smoking intensity between populations. The authors might want to consider nuancing this statement.

Response: The reviewer is absolutely right and agree with the reviewer that our concluding sentence was an overstatement. We have now modified that sentence to simply state that the top smoking-associated DMCs are largely independent of ethnicity/genotype. In other words, there could be lower-ranked DMCs that are dependent on ethnicity.

Comment: How do the authors explain the large discrepancy between the number of CpGs identified in the ordinary EWAS analysis in the TZH cohort (63977) versus the cell lineage-specific analysis (170 myleoid-specific and no lymphoid-specific). Could they comment on this in the discussion? Does this reflect the lower power of the CellDMC interaction model or does it mean that the majority of smoking-associated methylation signal is in fact NOT lineage-specific? If a methylation difference is present in both the lymphoid and myeloid lineage, how does this present in CellDMC?

Response: We thank the reviewer for raising this excellent question. The answer comes in two parts. First, we acknowledge that the 63977 DMCs at FDR<0.05 as inferred in the TZH cohort may be an overestimate. Even if we were to use a Bonferroni threshold, we would still have 5308 smoking-DMCs in the TZH cohort. We note that although the recent large meta-analysis study of Joehanes et al (now cited and discussed in the revised manuscript) discovered at Bonferroni significance on the order of 2600 smoking-DMCs, we feel that this may also be an overestimate. Indeed, we note that over 50% of the presumed 2600 gold-standard smoking-DMCs from Joehanes et al did not validate in the TZH cohort (this is now mentioned in the main text and related data is shown in a new Fig.1b), so we suspect that the true number of smoking-associated DMCs is lower, and probably not more than 1000. We note that inflation is an ubiquitous feature of these studies, as it is extremely difficult to avoid the effect of hidden unknown confounders. An unknown study-specific confounder could

easily inflate statistical significance estimates, even in the context of a meta-analysis study. We further note that the study by Gao & Brenner (Clin Epigenet 2015) who performed a review of 12 whole blood smoking-EWAS, only arrived at a relatively small number of gold-standard smoking-DMCs: approximately 151 smoking-DMCs were observed in at least 2 studies, and 62 in at least 3 of the 12 studies. Thus, although it is clear that individual studies may report on the order of thousands of DMCs, when we assess the consistency across independent studies we arrive at a much smaller number. We further note that almost all of these 62 CpGs were significant in the TZH cohort (as shown in Fig.1a), which if compared with the less than 50% that validated from the larger list of ~2600 smoking-DMCs from Joehanes et al, suggests that indeed the number of true smoking-DMCs (as assessed in whole blood) is on the order of hundreds. In the TZH cohort itself, one potential confounder which could explain the inflation, but which we are unable to adjust for, is sex. Indeed, because in this Chinese cohort over 95% of smokers are males (with the proportion of males in non-smokers being only 27%) adjusting for sex would deflate and effectively remove the well-known associations with smoking (e.g. many of the AHRR hits would disappear if we were to adjust for sex, which would contradict a large body of literature). While we could have run the analysis in the TZH cohort using only males, this would however render the study less powerful to detect associations with smoking, most of which are probably also sex-independent. Indeed, as far as the reported gold-standard lists of smoking-associated DMCs is concerned, there is not much evidence that these are sex-specific.

Now to the second part of our answer: even if the true number of smoking-DMCs in whole blood is around 1000, when running CellDMC to find cell-type specific DMCs (i.e. DMCTs), we still observe a substantial drop in the number of hits, as correctly pointed out by the reviewer. Addressing this issue is complex because for most loci it is not known in which cell-types they are changing. While we provided sensitivity measures for a wide range of different scenarios in our CellDMC paper (see Zheng SC et al Nat Methods 2018), we have now conducted a more focused simulation in order to explain the above phenomenon. Smoking-DMCTs could be of 3 types: (i) they change in both myeloid and lymphoid lineages (non-specific DMCTs), (ii) they only change in the myeloid lineage (myeloid-specific DMCTs), (iii) they only change in the lymphoid lineage (lymphoid-specific DMCTs). We have run simulations for the 3 different scenarios for realistic sample sizes, cell-type fractions (myeloid fraction is higher than lymphoid fraction) and effect sizes, and the results of these simulations (shown in a new Figure-5) indicate that we have adequate power to detect lymphoid-specific or myeloid-specific DMCTs if these exist. However, the simulations also demonstrate that the power to detect DMCTs within the lymphoid lineage is compromised if these same loci are also changing in the myeloid-lineage. This happens because the myeloid fraction is higher.

To summarize, we believe that the large drop in hits when going from DMCs to DMCTs owes mainly to inflation at the level of DMCs. Although CellDMC appears to

have adequate power to detect lineage-specific DMCTs, power estimates depend on the effect sizes within individual cell-types, which are largely unknown. Running CellDMC on datasets profiling thousands of samples together may help shed further light on this question. In response to the reviewer's point, we have a new subsection in Results discussing the results of our simulation, and a new Fig.5 displaying the power calculation results of this simulation, and have also expanded Discussion to better interpret our results in the light of the simulation analysis.

Comment: Page 6 "Of the 60 hypomethylated gsSMK-CpGs, 53 passed QC". It is unclear to me why this is mentioned here, since the gsSMK-CpGs were also already discussed on page 4 (fig1A).

Response: We agree that we could have mentioned it earlier, but we felt that it would be better to emphasize the fact that not all 62 gold-standard CpGs were analyzed in relation to cell-type specific changes. We think that perhaps some of the confusion has arisen because we wrongly implied that QC removed a number of gold-standard CpGs. In fact, this is not the case. Of the 62 gold-standard CpGs, only 55 are present on the EPIC beadarray (recall that the gold-standard CpGs were derived from 450k arrays). Of these 55, 54 are hypomethylated according to Gao & Brenner, and 1 is hypermethylated. So, we apologize for wrongly implying that QC removed a number of gold-standard CpGs. In fact, only 1 of the 54 gold-standard hypomethylated CpGs was removed due to QC issues, resulting in 53 gold-standard hypomethylated CpGs and 1 hypermethylated one. We have now corrected the error and also improved the clarity.

Comment: Page 6 "and a substantial fraction of these exhibited hypomethylation in the myeloid..". could the authors be more specific (e.g. give the exact percentage in this sentence)

Response: In response to this, we now state that this fraction is 62%. Of course, the precise fraction depends on what is a somewhat arbitrary threshold for statistical significance, but the fraction of 62% is in line with the nominal $P < 0.05$ threshold as indicated in old Fig.1c (now new Fig.1d).

Comment: Page 6 "A similar trend was observed in the lymphoid cells, albeit not as strong as in the myeloid compartment". At this point, it is unclear what is meant with 'similar trend' (although this does become clear to me later in this paragraph). For clarity, could similar trend be replaced by something like 'showed the same direction of effect'?

Response: In response to this, we have altered the text in line with the reviewer's suggestion.

Comment: Maybe I missed it, but could the total sample size of the meta-analysis of 7 cohorts be presented somewhere?

Response: At the very top of the last paragraph in the Introduction we stated “Here, we apply CellDMC to 7 independent large EWAS cohorts, totaling 4448 samples,.....”

Comment: Page 13 “approximately 70 CpGs” sounds vague. Could the exact number be stated?

Response: We have now corrected this, stating that the signature comprises 69 CpGs.

Comment: Page 14 (discussion). The authors mention that it will be interesting to apply CellDMC to thousands of samples merged together in one dataset. If this would be possible, I wonder why they chose to apply a meta-analysis approach in the current paper?

Response: We thank the reviewer for raising this good point. Merging data from different cohorts together and then performing CellDMC is indeed an alternative strategy that we however did not pursue here, because it would require additional independent datasets for validation. This is because merging data from different cohorts together is tricky and while we can adjust for batch effects, linear adjustment does not adjust for potential differences in the variance. Thus, any results derived from a merged set, even after linear batch correction, would still in our opinion require validation in external data. It is therefore simpler and also more natural to first perform a meta-analysis over the independent studies, and to this end we adapted a powerful empirical meta-analysis method from Efron. With the meta-analysis approach we can determine overall statistical significance levels more reliably without the need for independent cohorts.

To perform the analysis on a merged set is a complex task we are currently addressing as part of a separate paper, as it requires correction for differences in variance between cohorts, so we are currently exploring novel methodology to achieve this.

Comment: Methods, page 15 “common SNPs were removed using minfi”. Which reference population did the authors use (is the common SNP filter in minfi suited for Asian populations?)?

Response: This is a good point. We used the package minfi to drop probes with SNPs and this package uses the common SNP tables from Illumina and UCSC. So, we acknowledge that we may have filtered out probes with SNPs which may not exhibit variation in a Chinese population, and conversely we may have included probes in our analysis that have SNPs in the TZH cohort. To assess whether this is a problem we have used the SNP database from the CONVERGE consortium, which has generated

a database of over 20 million SNPs from 11,670 genomes of Han Chinese (see *Cai N et al Sci Data 2017*). We were able to find 7582 SNPs with a MAF>0.004 (this corresponds to variation in about 3 of the 688 TZH samples with smoking information) that mapped to the interrogated CpG or the Single-Base Extension. The fraction of smoking-DMCs at FDR<0.05 which overlapped with any of these “SNP-probes” was 0.007. Using Bonferroni, there were 5308 smoking-DMCs, of which 40 overlapped with SNP-probes, so again a fraction of 40/5308 ~ 0.007, i.e. less than 1% of our probes may have been affected by SNP variation. As far as the CellDMC analysis is concerned, none of the identified DMCTs overlapped with these SNP-probes.

Comment: Software availability. The CellDMC tool is a highly valuable tool for the research community and it is great that this software package is freely available. Will the authors also make their script to perform meta-analysis of cell-type specific EWAS test statistics available? This would be highly valuable.

Response: We thank the reviewer for the encouraging feedback. We have made the meta-analysis R-script available as a function called “DoMetaEfron” within our EpiDISH BioC package (www.bioconductor.org/packages/devel/bioc/html/EpiDISH.html).

REVIEWER COMMENTS

Reviewer #1 (Remarks to the Author):

* The authors state there was no adjustment for sex in the TZH cohort due to sex*smoking differences. Examining supplement table 1. There appears to be heavy confounding by gender that should likely be addressed as the majority of never-smokers are females. Looking at the numbers, it would seem there is either 1 or 2 female former smokers and 3 or 4 female smokers??

This seems like an oversight. This analysis should likely only be run in Males.

* The authors say they removed sex chromosomes, but did they also remove cross-reactive probes?

* In the initial TZH analysis, the authors used the q-value, but in the later Meta-analysis they used Benjamini-Hochberg. Why the two different FDR correction methods?

* Tsaprouni study: would any unknown technical effects be picked up by PC's within the data

* The authors bring up pack-year, but can they discuss why they did not treat smoking status as categorical? Perhaps in the Cell-DMC results, the significant CpGs sites should be rerun as categorical with non-smokers as the reference.

* ZhangHIV450k/850k: There are no former smokers in this cohort. How was this accounted for? This seems it would not be comparable to the other studies

Line 384: Was a signature developed? It seems that a set of potentially interesting CpG sites were found that could maybe be followed up.

Minor

Line 84: "Although.... " bit of a run-on sentence.

Line 484: Missing instead of NA may be preferable

* For Hannum and Zhang HIV studies, was genetic ancestry accounted for.

Line 424: "Thus, either...." Run on sentence.

Reviewer #2 (Remarks to the Author):

The authors have largely addressed my original comments. I appreciate the additions in the Results and Discussion around ZEB2, RARA, RAD52, TELO2 and RPTOR, which help to connect the DNAm alterations to downstream health effects.

Overall I think the paper supports myeloid specificity of many smoking-related DNA methylation alterations. However, as described below, I have some additional detailed requests that I think the authors could satisfy without too much trouble.

(1) Regarding p-values (my original minor comment #1 and the authors' response), I understand that p and q value thresholds are largely a matter of social convention. However, given that q-

values already relax the restrictions implicit in the use of p-values, I wonder about the relaxation of q-value thresholds from 5% to 10% or 30%. In general, the authors are using these only as a matter of convenience to provide a definitive list of CpGs from one study to validate in another, so in the bigger picture it doesn't matter too much. I might have instead used a more agnostic approach such as Wilcoxon/ROC-AUC. My modest request to the authors is to (a) state up front the over-arching issue of convenience as a rationale for the various thresholds used and (b) where ORs and p-values from Fisher tests are given (e.g. Figure 1B), add ROC-AUC and the corresponding Wilcoxon p-value. These could simply be stated in a supplementary table if they make the figures too busy. I don't expect the scientific conclusions to change at all, but it removes the dependence of individual results on the choices of threshold, which the authors themselves admit is arbitrary.

(2) I have a comment regarding the new simulation results presented in this revised version. Lines 315-319 state the following: "Reducing the variance within case/control groups whilst keeping the average methylation difference at around 10% (corresponding to an effect size close to 2), revealed approximately 90% sensitivity to detect lymphoid-DMCTs (Fig.5a), but only if these are specific to the lymphoid-lineage (i.e. not altered in myeloid cells) (Fig.5b). These results support the view that lymphoid-specific DMCTs are rare." I may be missing something, but it seems that the results of the simulation also support the possibility that some of the putative myeloid-specific alterations could actually not be specific to the myeloid lineage (Figure 5c). Should the authors not acknowledge this in an additional sentence? I understand that some of the other results the authors have presented in this paper supports myeloid specificity, but the lack of myeloid specificity in any one individual analysis needs to be acknowledged. Alternatively/additionally, the authors might add another row of panels that show specificity in the same way that the existing rows show sensitivity. I believe these would be of interest anyway.

Reviewer #3 (Remarks to the Author):

The paper is well written and fill an important literature gap. I have no additional comment for the authors.

Reviewer #4 (Remarks to the Author):

I thank the authors for addressing my comments and answering my questions. I have no further comments.

Detailed Responses to Reviewer's comments:

Reviewer #1:

*Comment: The authors state there was no adjustment for sex in the TZH cohort due to sex*smoking differences. Examining supplement table 1. There appears to be heavy confounding by gender that should likely be addressed as the majority of never-smokers are females. Looking at the numbers, it would seem there is either 1 or 2 female former smokers and 3 or 4 female smokers?? This seems like an oversight. This analysis should likely only be run in Males.*

Response: We would like to thank the reviewer for once again taking time to evaluate our manuscript. We appreciate the reviewer's concern, but can assure him/her that there was no oversight and that we were fully aware of the fact that Chinese females are generally non-smokers. In fact, in Methods we had already stated in relation to the TZH cohort that "We did not adjust for sex, because in this cohort sex was very strongly correlated with smoking status. Indeed, if we were to include sex as a covariate, all associations with smoking would disappear including top hits like AHRR, plainly contradicting the well-known observation that the AHRR CpGs correlate with smoking in both males and females". However, in revising our paper we discovered an error in our previous analysis of DMCs (i.e. the data shown in Fig.1a-b). In this previous analysis, when we adjusted for sex we had not found any DMCs, but this was due to an unfortunate bug in the R-script. Correcting this bug and adjusting for sex, we do retrieve the well-known smoking-associated DNAm signature (e.g. AHRR), and reassuringly we no longer have the previous inflation, which we now believe was caused entirely by sex-effects. At an FDR < 0.05, we now find just over 400 DMCs, which is much more sensible than the 65,000DMCs we had reported previously. This new data is shown in new Fig.1a. The agreement with the DMC list from Joehanes et al is now even stronger (see new Fig.1b).

As far as the CellDMC analysis in the TZH cohort is concerned, we have now also adjusted for sex, and this has had a moderate effect on global significance levels, but otherwise all other analyses, including the results of the meta-analyses, remain largely unchanged. Some of the numbers have changed, reflecting the modified analysis in the TZH cohort, but the main results and conclusions have not been altered.

Comment: The authors say they removed sex chromosomes, but did they also remove cross-reactive probes?

Response: We thank the reviewer for asking this and apologize for the lack of clarity. We did remove cross-reactive probes in the Meta-analysis, but did not do so in the first part of the paper that deals exclusively with the TZH cohort. In our opinion, the decision to exclude potential cross-reactive probes is very much dependent on the nature of the downstream analysis, as there are equally good arguments for keeping them in, or excluding them. The reason why they were removed in the Meta-analysis is that ultimately we perform a GSEA using eFORGE, and we need to ensure that we have no cross-reactive probes, as this could confound enrichment and biological interpretation. On the other hand, in the initial analysis performed in the TZH cohort,

we were mainly interested in deriving DMCs and DMCTs in a Chinese population, and any cross-reactive probes can be removed a-posteriori if required. Indeed, we note that other studies also adopt this inclusive approach. For instance, using a list of 46,751 cross-reactive Illumina (EPIC&450k) probes derived from three papers: Chen YA et al Epigenetics 2013, MacCartney CL et al Genomics Data 2016 and Pidsley R et al Genome Biol.2016, we have checked that among the gold-standard list of 2622 smoking-associated DMCs from Joehanes et al, that there are 85 cross-reactive probes. We note that these 2622 sites were identified from a meta-analysis over 16 cohorts and over 10,000 samples, and that therefore by removing these 85 probes we could potentially also lose valuable information. In the same way as Joehanes et al, we prefer an inclusive approach where we keep potential cross-reactive probes in the analysis, and only remove them when followed-up in e.g. a GSEA-type of analysis. In response to the reviewer's point, we have now clarified in Methods that cross-reactive probes were removed in the Meta-analysis.

Comment: In the initial TZH analysis, the authors used the q-value, but in the later Meta-analysis they used Benjamini-Hochberg. Why the two different FDR correction methods?

Response: We thank the reviewer for raising this technical point. In principle, the q-value approach to FDR estimation is preferable, however, it is also much more sensitive and can break down in cases where the true null distribution is not well described by the analytical one. This can easily happen when performing meta-analyses, which is why the Benjamini-Hochberg procedure is often used in this context (see e.g. the EAMA procedure in Sikdar S et al PLoS One 2017). To ease any potential concern the reviewer may have, in the revised version we have decided to also estimate the FDR via an empirical Monte-Carlo randomization procedure, i.e. we derive an empirical null, and the obtained results are similar to those using Benjamini-Hochberg. This is described in the Methods section and we have also added a new panel a) to Fig.4, to show the results of this empirical FDR estimation method.

Comment: Tsaprouni study: would any unknown technical effects be picked up by PC's within the data

Response: The reviewer has raised a valid point, but unfortunately the study by Tsaprouni et al did not provide batch information. The only information we have is age, sex, smoking status, and the cell-type fractions which we can estimate from the DNAm data itself. If we perform a SVD/PCA and ask what the PCs correlate with (see right panel in figure below), we find that the top component correlates most strongly with cell-type fraction and sex, that the 2nd component correlates with cell-type fractions and smoking, with lower ranked components correlating mostly with cell-type fractions and age. Technical factors such as beadchip/position/plate are likely to be present, but without knowledge of these factors we can't identify which specific PCs we should remove. If we were to include all top PCs as covariates, danger is we could be removing the effect of smoking. We note that since PC-1 is associated with sex and cell-type fractions, and since sex and cell-type fractions were included as covariates

in our models, that we are effectively adjusting for PC-1. By also adjusting for age, we are definitely adjusting for the major sources of confounding variation, which as the left panel shows drive most of the variation in the data-matrix. In our experience, given the nature of the dominant variation in this dataset, we think that the adjustment strategy we have used is sensible.

Comment: The authors bring up pack-year, but can they discuss why they did not treat smoking status as categorical? Perhaps in the Cell-DMC results, the significant CpGs sites should be rerun as categorical with non-smokers as the reference.

Response: We thank the reviewer for raising this point. In our opinion, it is clear from the literature on smoking-associated DNAm changes (see Tsaprouni et al Epigenetics 2014, Teschendorff et al JAMA Oncology 2015) that DNAm reverts back to a normal state when ex-smokers have quit smoking for more than 10-years before the blood sample was taken. Thus, it is reasonable to treat the category “ex-smokers” as representing an intermediate level of exposure, and therefore to treat never-smokers, ex-smokers and smokers as an ordinal i.e. (0,1,2) variable. There are a number of reasons why we would not want to treat never-smokers, ex-smokers and smokers as categorical, i.e. treating them as apples, oranges and bananas. First of all, our paper is about trying to assess the likely cell-lineage in which smoking-associated DNAm changes happen, and in particular we are interested in assessing this for the well-known smoking-associated DNAm loci (e.g. AHRR) which do show partial reversal upon smoking cessation. It therefore makes sense to use a linear model with smoking as an ordinal numerical variable. Second, the CellDMC algorithm that we proposed back in 2018 has only been tested and validated in the context of numerical/ordinal exposures. A third related reason is that CellDMC requires as large a sample size as is possible, otherwise power could be compromised. By using smoking status as an ordinal variable and a corresponding linear model to identify DMCTs, power to detect such linear changes is maximized. Fourth, we are not interested in DNAm changes in ex-smokers which are not seen in current-smokers. Finding these type of non-monotonic changes would be the main reason for treating smoking as categorical, and yet their interpretation would also be very difficult and are more likely to represent false positives arising from residual confounding due to imprecise epidemiological variables. Fifth, adopting a linear ordinal model has very similar power than a categorical one to find DNAm changes that only occur in smokers, i.e. CpGs that are indistinguishable

between never and ex-smokers, but which differ in relation to smokers. In summary, treating smoking-status as an ordinal variable is in our opinion the most sensible approach given the need to maximize power within a statistical model that includes interaction terms (ie CellDMC).

Comment: ZhangHIV450k/850k: There are no former smokers in this cohort. How was this accounted for? This seems it would not be comparable to the other studies

Response: We don't understand why this is a concern. Once again, it is sensible to treat ex-smokers as an intermediate category, so if this intermediate category is missing, one can still identify changes between the two extremes. It would be more justified to drop a study if it did not have non-smokers or current smokers, as the extremes in the phenotype are important. Finally, we should also point out that it is not the purpose of this manuscript to investigate a "former-smoker specific" smoking-associated signature.

Comment: Line 384: Was a signature developed? It seems that a set of potentially interesting CpG sites were found that could maybe be followed up.

Response: Line 384 refers to our simulation/power analysis, so the answer here is that no signature was developed. If the reviewer meant line 348, we provide a list of the enriched CpGs mapping to DHSs in AML in Supplementary Data 6. The suggestion to follow these up is indeed an excellent one, but we view this as a longer-term project, which requires very special datasets to explore further. We are currently also pursuing the alternative strategy to merge DNAm datasets first, and then apply CellDMC to see if we can improve the resolution of this AML signature.

Minor points:

**Line 84: "Although.... " bit of a run-on sentence.*

Response: We prefer to leave the decision to change this to the editor, as we feel that the current expression is reasonably good English.

**Line 484: Missing instead of NA may be preferable*

Response: We have changed one of the NAs to missing, as requested.

** For Hannum and Zhang HIV studies, was genetic ancestry accounted for.*

Response: As already explained in Methods, in the Hannum et al dataset race was distributed in a plate-specific manner, and so adjusting for plate automatically adjusts for race too. In the case of the Zhang HIV studies, adjusting for ethnicity did not substantially alter the statistics, as shown in the figure below, where the CellDMC statistics without adjustment for ethnicity are shown along x-axis, whilst those adjusted for ethnicity are shown on y-axis.

*Line 424: "Thus, either...." Run on sentence.

Response: We have modified this.

Reviewer #2:

General comment: The authors have largely addressed my original comments. I appreciate the additions in the Results and Discussion around ZEB2, RARA, RAD52, TELO2 and RPTOR, which help to connect the DNAm alterations to downstream health effects.

Response: We thank the reviewer for re-evaluating our manuscript and for the positive feedback provided.

Comment (1): Regarding p-values (my original minor comment #1 and the authors' response), I understand that p and q value thresholds are largely a matter of social convention. However, given that q-values already relax the restrictions implicit in the use of p-values, I wonder about the relaxation of q-value thresholds from 5% to 10% or 30%. In general, the authors are using these only as a matter of convenience to provide a definitive list of CpGs from one study to validate in another, so in the bigger picture it doesn't matter too much. I might have instead used a more agnostic approach such as Wilcoxon/ROC-AUC. My modest request to the authors is to (a) state up front the over-arching issue of convenience as a rationale for the various thresholds used and (b) where ORs and p-values from Fisher tests are given (e.g. Figure 1B), add ROC-AUC and the corresponding Wilcoxon p-value. These could simply be stated in a supplementary table if they make the figures too busy. I don't expect the scientific conclusions to change at all, but it removes the dependence of individual results on the choices of threshold, which the authors themselves admit is arbitrary.

Response: We thank the reviewer for raising these important issues. (a) If the minimum FDR over all features is 1, this means that there is no global association. Let us assume now that $\min(\text{FDR}) < 1$, and that there are say 100 features all with $\text{FDR} < 0.3$. This means that 70% of the 100 features are expected to be true positives, only 30% are false positives, i.e. a substantial majority of the 100 features are likely to be true

positives. While using an FDR < 0.3 threshold might seem unconventional, we have in the past been able to validate molecular signatures, that were derived at this FDR < 0.3 threshold in a discovery set, in completely independent datasets (see e.g. Naderi A, Teschendorff A et al Oncogene 2007). This should not be surprising if indeed 70% of the features are true positives. In our experience, an FDR < 0.3 threshold is as high as one can reasonably go when relaxing significance thresholds. Relaxing the FDR threshold from the usual 0.05 to 0.3 would also be advisable for instance when performing GSEA, in order to increase the number of selected genes (if this number is too low at FDR < 0.05), and thus to increase power. So, by relaxing a threshold to FDR < 0.3, means that we can capture more true positives to increase power in a GSEA, whilst the proportionate increase in false positives is lower. Importantly, the increased number of false positives is unlikely to cause major confounding in a GSEA, because by definition null-genes would map to any random pathway/biological term, and therefore false positives should not enrich for any specific pathway/biological term. In summary, all of this is well-known and discussing these technical details in the manuscript, as the reviewer is suggesting, would in our sincere opinion only disrupt the natural flow and be a distraction.

(b) We appreciate the reviewer's point but unfortunately don't understand why this reviewer wants us to use ROC/AUC in this context. In fact, the purpose of a ROC/AUC is to assess sensitivity/specificity and overall classification accuracy, but the sole purpose of Fig.1b is to demonstrate that the DMCTs listed in Joehanes exhibit the same directional changes in our TZH cohort. Fig.1b is not about classification accuracy. It is extremely clear from Fig.1b that the association is highly significant, and so what is the purpose of cluttering this panel with an AUC value that most readers would find confusing and uninformative. We are great fans of the Wilcoxon-test and the AUC, but in our opinion the scatterplot is a more powerful graphical way to display the agreement of the CpG statistics in the two studies, and the OR and Fisher-test P-value are the more appropriate statistics to use that are matched to this type of figure.

Comment (2): I have a comment regarding the new simulation results presented in this revised version. Lines 315-319 state the following: "Reducing the variance within case/control groups whilst keeping the average methylation difference at around 10% (corresponding to an effect size close to 2), revealed approximately 90% sensitivity to detect lymphoid-DMCTs (Fig.5a), but only if these are specific to the lymphoid-lineage (i.e. not altered in myeloid cells) (Fig.5b). These results support the view that lymphoid-specific DMCTs are rare." I may be missing something, but it seems that the results of the simulation also support the possibility that some of the putative myeloid-specific alterations could actually not be specific to the myeloid lineage (Figure 5c). Should the authors not acknowledge this in an additional sentence? I understand that some of the other results the authors have presented in this paper supports myeloid specificity, but the lack of myeloid specificity in any one individual analysis needs to be acknowledged. Alternatively/additionally, the authors might add another row of panels that show specificity in the same way that the existing rows show sensitivity. I believe these would

be of interest anyway.

Response: In relation to the interpretation above, the reviewer is absolutely correct, and we apologize if this had not been clearly spelled out. We note that we had stated in the Results section that “...., for DMCTs occurring simultaneously in both lineages, the sensitivity to detect them in the lymphoid lineage is severely compromised, while the sensitivity to detect them in the myeloid lineage remains over 50% (Fig.5b)”. We end that paragraph by further stating that “These results support the view that lymphoid-specific DMCTs are rare”. This means that there could be smoking-associated DNAm changes in lymphoid cells but that these would also occur simultaneously in myeloid cells. In response to the reviewer’s point we have now added an additional sentence for clarification. We note that to further resolve this question, will require larger datasets and perhaps also alternative strategies such as merging of datasets before running CellDMC. Indeed, we are currently pursuing the latter strategy on larger datasets.

As far as the second suggestion is concerned, we worry that the reviewer is confusing the term “specificity”, as this can mean different things depending on context. To clarify, our simulation figure is aimed at addressing the question as to why we don’t find many lymphoid-DMCTs, and therefore it is aimed at quantifying sensitivity, which is indeed the relevant metric. Normally, when we estimate sensitivity to detect DMCTs, one can also estimate the precision, i.e. the fraction of true positive calls among all significant calls, and the specificity defined as $1 - \text{FPR}$ (false positive rate), where the FPR quantifies the fraction of false positives among all true non-DMCTs. The specificity and PPV for our simulation model is very close to 1 throughout the whole range of effect sizes, which is why we did not display it. The high specificity and PPV exhibited by CellDMC is not surprising and is consistent with the data shown in our earlier publication (see Zheng SC et al Nat Methods 2018). In other words, when running CellDMC with sensible significance thresholds, the main limitation is sensitivity, not specificity or precision. Of course, when applied to real data, unknown confounders can complicate matters, but generally speaking sensitivity is the most important metric. In response to the reviewer’s point we have added 4 new panels c-f) to Fig.5, which display the specificity and PPV, and the sensitivity for two choices of total sample size (n=200 & n=600), which for convenience we display again below (see also new Suppl.Fig.S8):

Reviewer #3:

Comment: The paper is well written and fill an important literature gap. I have no additional comment for the authors.

Response: We sincerely thank the reviewer for taking time to evaluate our manuscript once again, and for the positive feedback.

Reviewer #4:

Comment: I thank the authors for addressing my comments and answering my questions. I have no further comments.

Response: We sincerely thank the reviewer for taking time to evaluate our manuscript once again, and for the positive feedback.

REVIEWERS' COMMENTS:

Reviewer #1 (Remarks to the Author):

1) The methods section still says that there was no adjustment for sex in TZH cohort and that doing so erases all signal. Please update.

2) Why the different missing rate cutoffs between TZH (.05) and ZhangHIV(.01)?

3) Would a two-df test have revealed more potential DMCTs? I.e. $H_0: \beta_{c,mye} = \beta_{c,lymph} = 0$ vs H_1 : at least one $\beta_{c,.}$ is not zero. (no interactive effect, yes interactive effect). Is the reason as it may lead to issues in the meta-analysis?

No further comments.

Reviewer #2 (Remarks to the Author):

I don't think the authors understood my suggestion about ROC/AUC: when comparing p or q values to a dichotomous variable, if a definitive significance threshold is not convenient for comparison, it can be helpful to use a Wilcoxon test to compare p/q-values across the two categories. The corresponding measure of association strength would be AUC. However, it wasn't that important in the bigger picture, and I don't think this detail warrants holding up this manuscript any further.

I thank the authors for addressing/clarifying my other points. As far as I'm concerned, the paper is publishable in its current form.

Detailed responses to Reviewer #1:

Comment 1): The methods section still says that there was no adjustment for sex in TZH cohort and that doing so erases all signal. Please update.

Response: We apologize for forgetting to correct this, and thank the referee for pointing this out to us. We have now updated this.

Comment: 2) Why the different missing rate cutoffs between TZH (.05) and ZhangHIV(.01)?

Response: Yes, the reviewer is right that we used slightly different P-value thresholds for calling NAs, but as stated in Methods we used 0.01 for TZH and 0.05 for ZhangHIV, which is opposite to what the reviewer is stating. The choice of P-value threshold is motivated by a whole plethora of factors, including the maximum number of NAs that can be reliably imputed without incurring substantial bias, and the number of probes to be retained. Although generally speaking we tend to use a $P < 0.05$ threshold for calling significant measurements above background, in this study we decided to use a slightly more stringent threshold ($P < 0.01$) in the TZH cohort. This is because the overall coverage per sample in the TZH cohort was much higher than in other studies, allowing for a more stringent detection threshold to be used without compromising on total probe number and imputation accuracy. We have now added one sentence to Methods to explain why we used a more stringent $P < 0.01$ threshold in the TZH cohort.

Comment 3) Would a two-df test have revealed more potential DMCTs? I.e. $H_0: \beta_{c,mye} = \beta_{c,lymph} = 0$ vs $H_1: \text{at least one } \beta_{c,.} \text{ is not zero. (no interactive effect, yes interactive effect). Is the reason as it may lead to issues in the meta-analysis?}$

Response: When running CellDMC with only 2 cell-types, there is a linear dependency between the two fractions, since $f_{lym} + f_{mye} = 1$. So, we don't think that the suggested model is any different to what we have run. In other words, we can run an equivalent model with only 1 interaction term (for one cell-type) and which then includes a linear non-interaction term in y . In any case, we feel that the reviewer's fundamental concern is fully addressed in our simulation analysis shown in Fig.5. What the sensitivity curves in Fig.5a-b demonstrate is that we do have enough power to detect myeloid and lymphoid specific DMCTs if these exist. However, for non-specific DMCTs (ie those that happen in both lineages) detecting them in the lymphoid lineage may be hard depending on effect size, as panel Fig.5b demonstrates. In summary, our data suggests that the well-known smoking associated DMCTs reported in many previous EWAS are more prominent in the myeloid lineage, and that there are few lymphoid-specific smoking DMCTs.